## Special feature reviews

biomechanics, biophysics, behaviour

locomotion, obstacle traversal, self-righting, robophysics, terradynamics, neuromechanics

**Author for correspondence:**
Chen Li
e-mail: chen.li@jhu.edu

Stability and manoeuvrability in animal movement: lessons from biology, modelling and robotics. Guest edited by Andrew Biewener, Richard Bomphrey, Monica Daley and Auke Ijspeert.

# Locomotor transitions in the potential energy landscape-dominated regime

Ratan Othayoth, Qihan Xuan, Yaqing Wang and Chen Li

Department of Mechanical Engineering, Johns Hopkins University, 3400 N. Charles Street, Baltimore, MD 21218, USA

RO, 0000-0001-5431-9007; QX, 0000-0002-1075-1516; YW, 0000-0001-6884-323X; CL, 0000-0001-7516-3646

To traverse complex three-dimensional terrain with large obstacles, animals and robots must transition across different modes. However, most mechanistic understanding of terrestrial locomotion concerns how to generate and stabilize near-steady-state, single-mode locomotion (e.g. walk, run). We know little about how to use physical interaction to make robust locomotor transitions. Here, we review our progress towards filling this gap by discovering terradynamic principles of multi-legged locomotor transitions, using simplified model systems representing distinct challenges in complex three-dimensional terrain. Remarkably, general physical principles emerge across diverse model systems, by modelling locomotor–terrain interaction using a potential energy landscape approach. The animal and robots' stereotyped locomotor modes are constrained by physical interaction. Locomotor transitions are stochastic, destabilizing, barrier-crossing transitions on the landscape. They can be induced by feed-forward self-propulsion and are facilitated by feedback-controlled active adjustment. General physical principles and strategies from our systematic studies already advanced robot performance in simple model systems. Efforts remain to better understand the intelligence aspect of locomotor transitions and how to compose larger-scale potential energy landscapes of complex three-dimensional terrains from simple landscapes of abstracted challenges. This will elucidate how the neuromechanical control system mediates physical interaction to generate multi-pathway locomotor transitions and lead to advancements in biology, physics, robotics and dynamical systems theory.

## 1. Introduction

To move about, animals can use many modes of locomotion (e.g. walk, run, crawl, slither, burrow, climb, jump, fly and swim) [1,2] and often transition between them [3,4]. Despite this multi-modality, the most mechanistic understanding of terrestrial locomotion has been on how animals generate [5–8] and stabilize [9–11] steady-state, limit cycle-like locomotion using a single mode.

Previous studies began to reveal how terrestrial animals stochastically transition across locomotor modes in complex environments. Locomotor transitions, like other animal behaviour, emerge from multi-scale interactions of the animal and environment across the neural, postural, navigational and ecological levels [12–14]. At the neural level, terrestrial animals use central pattern generators [15] and sensory information [16–18] to switch locomotor modes to traverse different media or overcome obstacles. At the ecological level, animals foraging across natural landscapes switch locomotor modes to minimize metabolic cost [19]. At the intermediate level, terrestrial animals transition between walking and running to save energy [20]. However, there remains a knowledge gap in how locomotor transitions in complex three-dimensional terrain emerge from physical interaction (i.e. terradynamics [21]) of an animal's body and appendages with the environment mediated by the nervous system. We lack theoretical concepts for thinking about how to generate and control locomotor transitions on the same level of limit cycles for steady-state, single-mode locomotion [22].

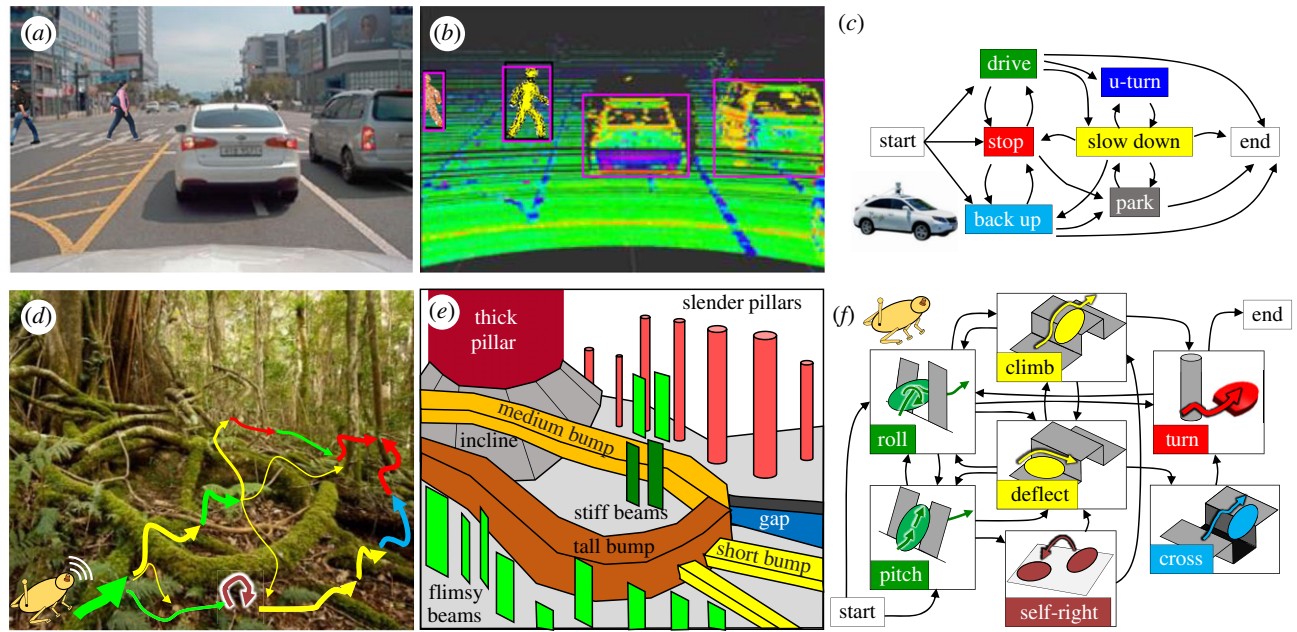

**Figure 1.** Multi-pathway transitions to avoid and traverse obstacles. (*a*) View from a self-driving car. (*b*) Geometric map scanned. (*c*) Multi-pathway driving transitions to avoid obstacles. (*d*) Envisioned capability of robot traversing complex three-dimensional terrain with many obstacles as large as itself. (*e*) Abstracted challenges from diverse large obstacles. (*f*) Envisioned multi-pathway locomotor transitions. Image credits: (*a,b*), Modified with permission from [23] under Creative Commons CC-BY license. (*d*) Modified with permission from Luke Casey Photography. (Online version in colour.)

Understanding of how to use physical interaction with complex three-dimensional terrain to generate and control locomotor transitions is also critical to advancing mobile robotics. Similar to personal computers in the 1970s, mobile robots are on the verge of becoming a major part of society. Wheeled robots like robotic vacuums and self-driving cars (figure 1*a*) already excel at avoiding sparse obstacles to navigate flat homes, streets and even unpaved roads, by scanning a geometric map of the environment (figure 1*b*) and acting upon it to transition between driving modes (figure 1*c*) [24]. This owes to the well-understood wheel–ground interaction physics [25,26]. Understanding of appropriate leg–ground physical interaction to generate and stabilize steady-state running and walking [5,6] enabled animal-like legged robot locomotion (such as from Boston Dynamics) on near-flat surfaces with small obstacles. However, despite progress in robot design, actuation and control for multi-modal locomotion [3], robots still struggle to make robust locomotor transitions to traverse obstacles as large as themselves, hindering important applications such as environmental monitoring in forests (figure 1*d*), search and rescue in rubble and extraterrestrial exploration through rocks. This is largely due to a poor understanding of physical interaction in complex three-dimensional terrain.

A physics-based approach by creating a new field of terradynamics [21] holds promise for filling this major gap. For aerial and aquatic locomotion of animals and robots, we understand fairly well their fluid–structure interaction thanks to well-established experimental, theoretical and computational tools, such as wind tunnel and water channel, aerofoil and hydrofoil, aero- and hydrodynamic theories, and computational fluid dynamics techniques [27]. By creating controlled granular media testbeds, robotic physical models [28,29], and theoretical and computational models, recent studies elucidated how animals (and how robots should) use physical interaction with granular media to move effectively both on and within sandy terrain (see [30] for a review). The general physical principles [30] and predictive physics models [21,30] not only advanced understanding of functional morphology [31–33], muscular

control [34,35] and evolution [36] of animals, but also led to new design and control strategies [28,30,37–40] that enabled a diversity of robots to traverse granular environments.

Inspired by these successes, our group has been expanding the field of terradynamics to locomotion in complex three-dimensional terrain, by integrating biological experiments, robotic physical modelling and physics modelling (figure 2). Here, we review our approaches, progress and opportunities ahead. This review focuses on multi-legged locomotor transitions; for our work on limbless locomotion in three-dimensional terrain, see [42–47]. We studied the rainforest-dwelling discoid cockroach (figure 3*a*), which is exceptional at traversing complex three-dimensional terrain with diverse large obstacles such as vegetation, foliage, crevices and rocks [4]. Just like how understanding the aerodynamics of passive aerofoils provides a foundation for understanding flight control [60], we first focused on understanding passive mechanical interaction, which provides a foundation for understanding sensory feedback control (and the intelligence aspect of locomotor transitions in general). This is achieved by studying the animal in the rapid, bandwidth-limited escape [61] or emergency self-righting response and feed-forward-controlled robotic physical models. Although still at an early stage, our work begins to reveal general physical principles of locomotor transitions, which is remarkable considering that complex three-dimensional terrain is highly heterogeneous with diverse obstacles. Our work again demonstrates the power of interdisciplinary integration to discover terradynamic principles.

## 2. Experimental tools

### (a) Model terrain

To begin to understand complex physical interaction during locomotion in nature (figure 1*d*), we abstracted complex three-dimensional terrain as a composition of diverse large obstacles (figure 1*e*) that present distinct locomotor challenges. These include compliant beams [50,51], rigid pillars [52], gaps [53] and bumps [54]. To enable systematic experiments (as in

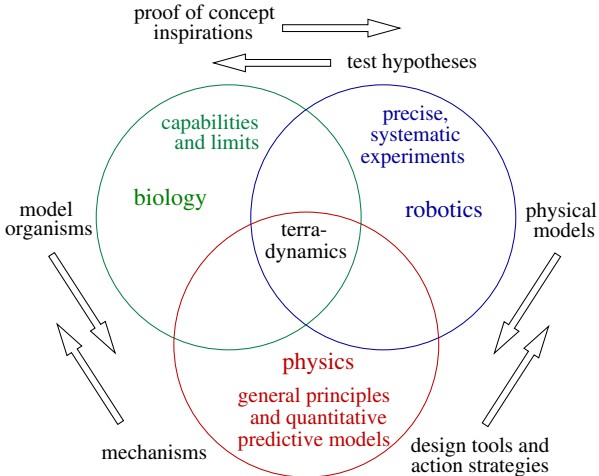

**Figure 2.** Integrative approach. Observations of model organisms inspire robot design and action. Simplified robots serve as physical models for testing biological hypotheses or generating new ones [28,29,41] and allow control and variation of parameters to discover general principles. Physical principles and predictive models from this empirical approach provide mechanistic explanations for animal locomotion and design tools and action strategies for robots. (Online version in colour.)

a wind or water tunnel), for each model terrain, we created a testbed that allowed controlled, systematic variation of obstacle properties such as stiffness [50], geometry [52] and size [53,54] (figure 3b). In addition, because animals and robots often flip over when traversing large obstacles [4,52,55], we studied strenuous ground self-righting in which existing appendages must be co-opted [55–59]. Furthermore, we developed tools to address technical challenges in measuring locomotor transitions and locomotor–terrain interaction in complex three-dimensional terrain (figure 3b–d; electronic supplementary material, Text S1).

Although studying locomotor transitions to overcome these challenges separately is an amenable first step (figure 1f), in the real world, animals and robots must continually transition across locomotor modes to traverse diverse obstacles over large spatio-temporal scales (figure 1e). To study continual transitions, we developed a terrain treadmill (figure 3e) to study locomotion through large obstacles over a long time and a large distance [48], while allowing finer features such as antenna and leg motion to be observed at a high spatial resolution [49]. This research direction is still at an early stage.

## (b) Robotic physical models

We created simplified robotic physical models [28,29] of each model system (figure 3f–j). These robots offer several advantages as experimental platforms. First, they generate relevant locomotor behaviour using minimalistic design, actuation and sensing, facilitating analysis and modelling. In addition, they are more amenable than animals to controlled parameter variation and hypothesis testing. Moreover, running the robot in open loop allows isolating the effects of passive mechanics from that of sensory feedback. Finally, they cannot violate the laws of physics because robots are enacting, not modelling, the laws of physics [62].

We emphasize that our robots were designed and controlled to generate relevant locomotor transitions that we studied, not optimized for maximal performance. However, the physical principles revealed by these tools are generalizable and can predict how to increase performance [4,28,29,50–55,57–59] (§4d).

# 3. Modelling approaches

## (a) Potential energy landscape modelling

Understanding how locomotor transitions emerge from locomotor–terrain interaction probabilistically (§4a) calls for a statistical physics approach. A statistical physics treatment has advanced understanding of complex, stochastic, macroscopic phenomena in self-propelled living systems, such as animal foraging [63], traffic [64] and active matter [65,66]. Here, we created potential energy landscape models (figure 4b), directly inspired by free energy landscapes for modelling multi-pathway protein folding transitions [67–69]. The near-equilibrium, microscopic proteins statistically transition from higher to lower, thermodynamically more favourable states on the free energy landscape. Thermal fluctuation comparable to free energy barriers induces probabilistic barrier crossings. These physical principles operating on a rugged landscape leads to multi-pathway protein folding transitions. Although our locomotor–terrain interaction systems are macroscopic, self-propelled and far-from-equilibrium, their locomotor transitions display similar features, including stochasticity, multi-pathway transitions, kinetic energy fluctuation (from oscillatory self-propulsion) and favourability of some modes over others [4,51–54,56–59], but with the addition of intelligence.

Given these similarities, we hypothesized that locomotor transitions are barrier-crossing transitions between basins of potential energy landscapes of our systems. We tested this hypothesis in each model system (figure 4; electronic supplementary material, text S3–S7). To discover general principles of locomotor transitions, we systematically varied system parameters and studied how they affect locomotor transitions. For how to use potential energy landscape modelling, see electronic supplementary material, text S2.

A potential energy landscape approach to modelling locomotor–terrain interaction is plausible also considering the success of potential energy field methods in modelling robotic manipulation. Similar to our systems, robotic part alignment [70] and grasping [71] have continual collisions, multiple pathways to reach the goal [70] and favourability of some contact configurations over others [72]. Given these complexities, quasi-static potential energy fields well explained how system properties like geometry and friction affect part-manipulator interaction and informed strategies to achieve desired alignment or manipulation [70].

We emphasize that our potential energy landscapes directly result from physical interaction and are based on first principles, unlike artificially defined potential functions to explain walk-to-run transition [73] and other non-equilibrium biological phase transitions [74], metabolic energy landscapes inferred from oxygen consumption measurements to explain behavioural switching of locomotor modes [19] and artificial potential fields for robot obstacle avoidance [75].

For simplicity, our potential energy landscapes so far only considered the most relevant system degrees of freedom (body rotation and translation in obstacle traversal, body rotation and wing opening in self-righting). In addition, they do not yet model system dynamics, which is required for the quantitative prediction of locomotor transitions (§5a). Despite these limitations, they provided substantial insight into the general principles and strategies of obstacle traversal and strenuous ground self-righting (§4).

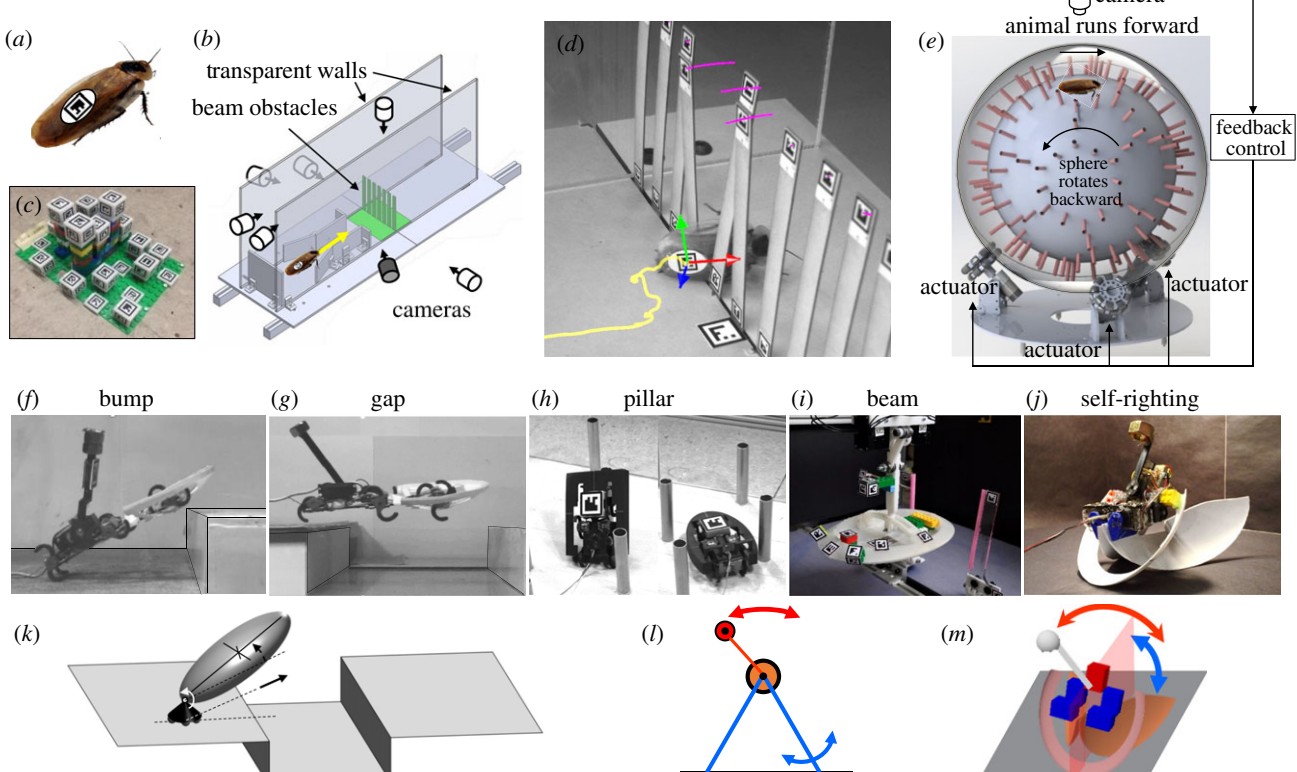

**Figure 3.** Experimental tools and dynamic models. (*a*) Model organism. (*b*) Terrain testbed with multi-camera imaging system. (*c*) Automated three-dimensional calibration object. (*d*) Snapshot of obstacle traversal showing automatically tracked trajectories of animal (yellow) and terrain (pink) markers. (*e*) Terrain treadmill with an untethered animal kept atop by rotating the sphere at the opposite velocity [48,49]. (*f–j*) Robotic physical models [4,50–59]. (*k,l*) Dynamical templates [53,58]. (*m*) Multi-body dynamics simulation [59]. (Online version in colour.)

## (b) Dynamic templates and simulations

Although our model systems follow Newton's laws, it is often challenging to solve equations of motion analytically due to the hybrid contact [22] and high-dimensional parameter space. As a first step to understand transition dynamics, we developed dynamical templates for two model systems, large gap traversal [53] (figure 3*k*) and strenuous ground self-righting [58] (figure 3*l*), for which equations of motion are solvable when two-dimensional dynamics is considered. Templates are the simplest dynamical models that capture the fundamental dynamics of a locomotor behaviour using minimal degrees of freedom [76]. For these two systems, our templates enabled quantitative prediction of contact and actuator forces [58], control strategies for traversal [53] or self-righting [58], and how they depend on system parameters [53,58].

In addition, for strenuous ground self-righting, we developed multi-body dynamics simulations of the robot validated against experiments [59] to study the effect of randomness in wing–leg coordination (figure 3*m*). These simulations enabled large-scale variation of relevant parameters identified from experiments and in-depth analysis at a precision difficult to achieve in animals and robots. Finally, simulation is faster than experiments [59].

# 4. Insights and general principles from simple model systems

Our studies revealed how locomotor transitions depend on system parameters (gap width, beam stiffness, body shape, etc.; electronic supplementary material, table S1). For each

model system, these physical principles are generalizable over the relevant parameter space and helped improve robot performance. Although our model systems are level, our approach also applies to interactions on slopes.

Across model systems, a potential energy landscape approach helps understand how the animal's and robot's stereotyped, probabilistic locomotor transitions are constrained by physical interaction. Several general physical principles and new concepts emerge.

## (a) Locomotor modes are stereotypical and transitions are stochastic

For all model systems, the animal displayed stereotyped locomotor modes with qualitatively similar body postural changes [4,50–54,56,57]. Not all modes lead to successful obstacle traversal or self-righting. Transitions between modes occur stochastically, with large trial-to-trial variation [4,50,51,53,54,56,57]. The probability of using or transitioning to a mode strongly depends on locomotor and terrain parameters that affect physical interaction [4,50,52–54,56,57] (§4f). The robot's locomotor modes are also stereotyped and transitions stochastic [4,50–55,57].

## (b) Locomotor transitions are destabilizing barrier-crossing transitions on a potential energy landscape

For all model systems, the system state in each mode is strongly attracted to a local minimum basin of the potential energy landscape over the relevant body state space [50–52,54,57] (figure 4; electronic supplementary material, figures S2–S6 and movie S1). This is because self-propulsion induces

royalsocietypublishing.org/journal/rspb　Proc. R. Soc. B 288: 20202734

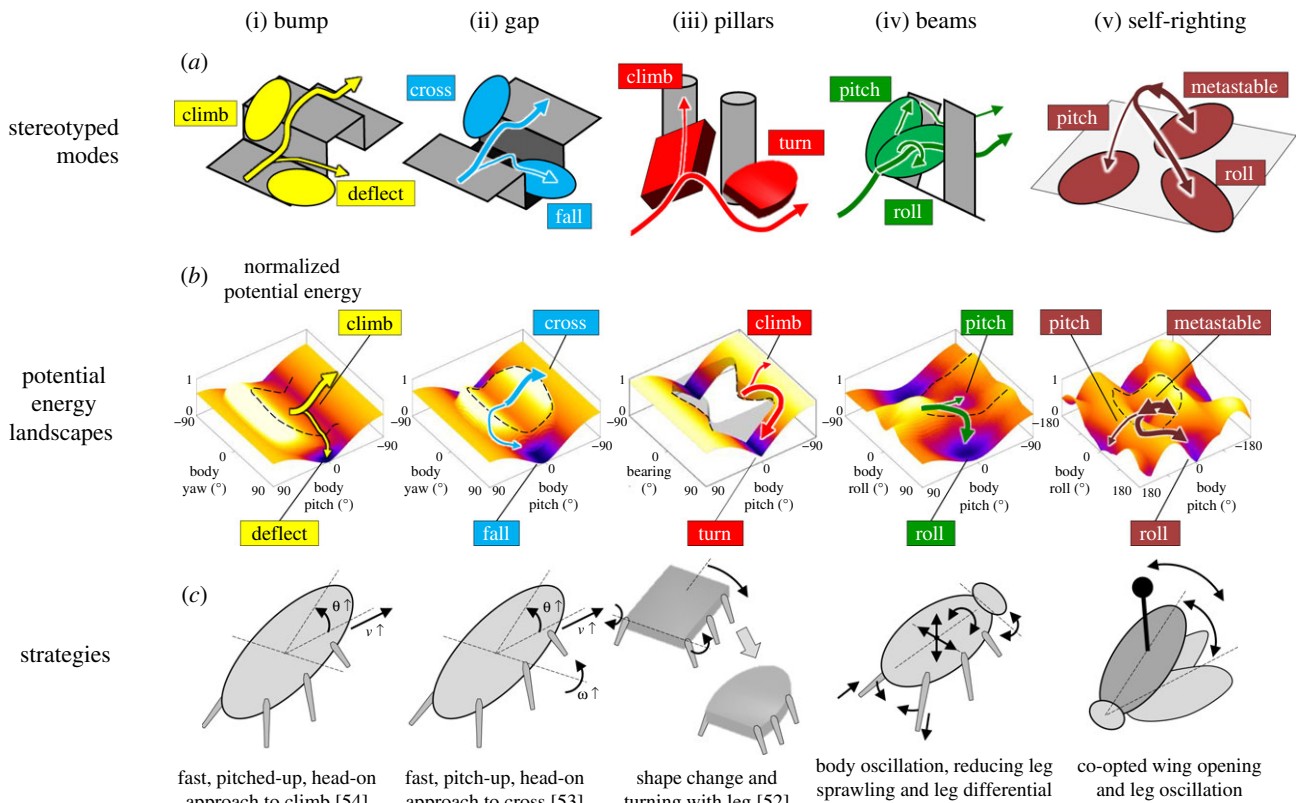

**Figure 4.** Modulation of locomotor transitions on potential energy landscapes via a suite of strategies. (a) Stereotyped locomotor modes of model systems. (b) Potential energy landscapes. The system is attracted to a distinct basin in each mode. A potential energy barrier must be crossed to make locomotor transitions. Black dashed curves show potential energy barriers. Arrows in (a,b) show representative system state trajectories; thicker arrows show more desirable modes. (c) Strategies that can increase the probabilities of desired modes and facilitate transitions to overcome locomotor challenges. (i–v) Model systems. See electronic supplementary material, table S1 and text S3–S7 for detail. We renamed some modes/basins in this review from in the original papers to better distinguish them across model systems. (Online version in colour.)

continual body–terrain collisions during obstacle interaction and self-righting, which breaks continuous frictional contact and makes the system statically unstable. This leads the system to drift down the basin until a sufficient perturbation induces an escape from the basin. However, the system does not stay at the minimum due to self-propulsion. Due to this strong attraction to landscape basins, the transition from one locomotor mode to another requires the system to destabilize itself to escape from one basin to fall into another.

## (c) There exists a potential energy landscape-dominated regime of locomotion

These observations across diverse model systems mean that there is a potential energy landscape-dominated regime of locomotion. In this regime, along with certain directions, there exist large potential energy barriers that are comparable to or exceed kinetic energy and/or mechanical work generated by each propulsive cycle or motion. This may happen when propulsive forces are either limited by physiological, morphological and environmental (e.g. low friction) constraints or are not well directed towards directions along which large barriers exist for the desired transition. These situations are frequent in large obstacle traversal and strenuous ground self-righting. In this regime, not only do potential energy landscapes provide a useful statistical physics approach for understanding locomotor transitions, but it also allows comparison across systems (different species [56], robots [4,52], terrain [50,52–54] and modes [4,50,52,56,57]) to discover general principles.

Outside of this regime, potential energy landscapes are not useful or necessary. Such examples include ballistic jumping over small obstacles with kinetic energy far exceeding potential energy barriers, moving on slopes with potential energy increasing or decreasing monotonically, and traversing obstacles much smaller or larger than body size.

## (d) Feed-forward self-propulsion can induce locomotor transitions

Using robotic physical models, we discovered several principles of locomotor transitions with feed-forward self-propulsion. First, locomotor kinetic energy fluctuation from self-propulsion helps the system stochastically cross potential energy barriers to make transitions [50,57]. In addition, escape from a basin is more likely in directions on the landscape along which the barriers are lower [50,57]. Finally, during a transition, the system tends to transition to more favourable modes attracted to lower basins [50,52,57]. The animal's locomotor transitions also largely followed these principles during rapid, bandwidth-limited escape or emergency self-righting response [50–54,56,57].

## (e) Feedback-controlled active adjustments can assist locomotor transitions

Not surprisingly, the animal can make active adjustments to facilitate or enable desired transitions when feed-forward self-propulsion is insufficient. For example, even when

body kinetic energy fluctuation becomes comparable to, but is still lower than, the potential energy barrier, the animal transitions to a more favourable mode to traverse beam obstacles [50], by actively adjusting body and appendages [51]. Understanding this intelligence aspect of locomotor transitions is clearly the next step. We have begun studying the principles of feedback-controlled locomotor transitions by creating robotic physical models with force sensing [51].

## (f) A suite of strategies can modulate locomotor transitions and increase performance

Because locomotor transitions are barrier-crossing transitions, they can be enhanced or suppressed by steering the system state on the landscape, changing landscape barriers, or even modifying landscape topology (the number of basins). This insight allowed us to discover a suite of strategies (figure 4c) to make desired transitions more probable for each model system (figure 4a), elaborated below.

In bump traversal, approaching with a head-on (body sagittal plane perpendicular to bump), pitched-up body posture directs the system to overcome a barrier to reach a desired climb basin/mode and avoid being attracted towards a deflect basin/mode (figure 4(i)) [54]. Similarly, in gap traversal, approaching with a large forward velocity and upward pitching velocity and a head-on, pitched-up body posture increases kinetic energy that directs the system to reach a desired cross basin/mode and avoid being attracted into a fall basin/mode (figure 4(ii)) [53].

In pillar traversal, a cuboidal body induces a climb basin/mode where the body is attracted to and pitches up against the pillar, whereas an elliptical body eliminates it and induces a desirable turn basin/mode where the body is repelled away (figure 4(iii)) [52]. Alternatively, active turning by legs helps a cuboidal body steer away from the climb basin/mode and cross the barrier to transition to the turn basin/mode [52]. In beam traversal, when beams are stiff, it is challenging to push across in a pitched-up mode attracted to a pitch basin, and it is desirable to transition to a roll mode/basin to roll into the beam gap to traverse (figure 4(iv)). Body kinetic energy fluctuation from self-propulsion helps cross the barrier to make this transition [50]. This transition is further facilitated by reducing sprawling and differential use of hind legs, which presumably destabilize and steer the system towards the roll basin [51].

In strenuous ground self-righting (figure 4(v)), although wing opening initiates a somersault and steers the system towards an upright pitch basin/mode, it is insufficient to overcome the large barrier. As a result, the system is frequently trapped in a metastable basin/mode due to a triangular base of support, leading to repeated failed attempts. However, wing opening reduces the barrier to transition from the metastable to a roll basin/mode, allowing small kinetic energy fluctuation from leg oscillation to induce barrier crossing, resulting in self-righting by rolling [57]. This transition is also facilitated by proper wing–leg coordination that better steers the system towards the lowered barrier to roll [58]. Randomness in wing–leg coordination helps find proper coordination [59].

We emphasize that the desirable modes and strategies in the obstacle interactions above aim at successful traversal. In different tasks, other modes may be desirable. For example, the fall mode in gap interaction (figure 4a,b(ii)) is desirable for going into ground crevices, and the climb mode for

pillar interaction (figure 4a,b(iii)) is desirable for initiating climbing up obstacles. Strategies can be discovered for these modes accordingly using the same approach.

Using our feed-forward-controlled robotic physical models [50,52,57–59] or with a human in the loop to switch on the strategies [52–54,58], we have demonstrated that these strategies increased robot performance substantially or even enabled new capabilities in each model system (electronic supplementary material, table S1). Efforts remain to study how robots should sense locomotor–terrain interaction and use feedback control to make transitions intelligently.

## (g) Stereotyped locomotor modes result from physical interaction constraint

Although the self-propelled system can in principle move in arbitrary ways, the observed locomotor modes are highly stereotyped due to strong constraints from physical interaction (§4a). This stereotypy is because the potential energy landscape is highly rugged, with distinct basins separated by barriers, and the system is strongly attracted to landscape basins in the potential energy landscape-dominated regime. Because our potential energy landscapes are directly derived from first principles (as opposed to fitting a model to behavioural data [77,78]), this insight provided evidence that behavioural stereotypy of animals emerges from the physical interaction of their neural and mechanical systems with the environment [12,13]. In addition, our systematic studies revealed that variation in movement can lead to stochastic locomotor transitions and is advantageous when locomotor behaviour is separated into distinct modes, each of which may be desirable for different scenarios.

We speculate that this physical constraint plays a role in the evolution of animal morphology and behaviour. This is plausible because morphological [79–81] and behavioural [82] adaptations that facilitate obstacle traversal and self-righting are common when microhabitat properties physically constrain movement. Our potential energy landscape approach is also useful for quantifying how physical interaction constrains robot design, control and planning for locomotor transitions in the large locomotor and terrain parameter space.

## (h) Physical principles of locomotor–terrain interaction are general

In the potential energy landscape-dominated regime, physical principles and strategies that we discovered (figure 4c; electronic supplementary material, table S1) are applicable to a broad range of the parameter space of model systems. For example, obstacle attraction or repulsion is an inherent property of the locomotor shape and insensitive to pillar size and geometry [52]. Strategies that favour bump or gap traversal are applicable to a large range of bump heights [54] or gaps widths [53]. Physical principles of beam interaction explained how pitch-to-roll transition probability changes as beam stiffness varies over a large range [50].

# 5. Towards multi-pathway locomotor transitions

Considering the general physical principles of locomotor transitions from diverse simple model systems, we hypothesize that multi-pathway locomotor transitions in heterogeneous

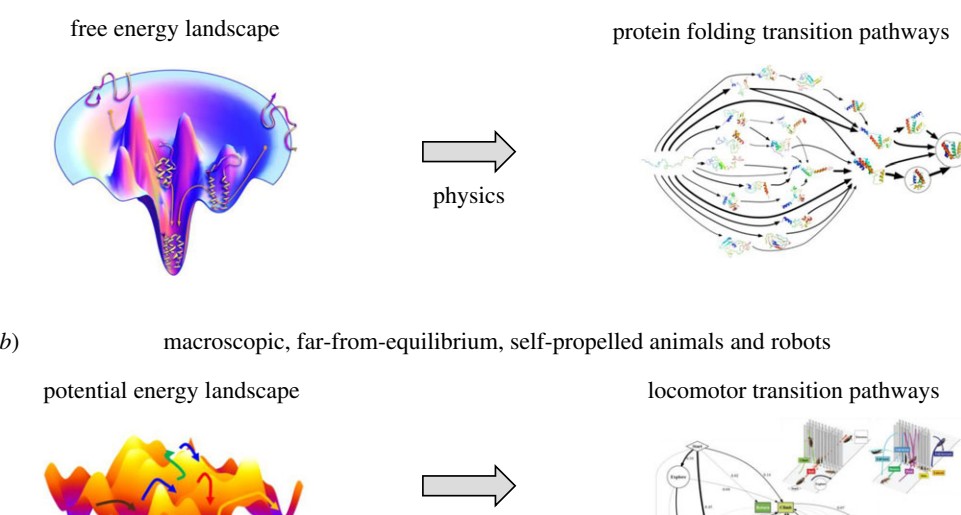

**Figure 5.** Comparison of two energy landscape approaches. (*a*) Rugged free energy landscapes help understand how proteins fold to their native states by stochastically transitioning from higher to lower free energy states via multiple pathways [67–69]. (*b*) We envision energy landscape modeling as a beginning of a statistical physics approach, but with the addition of intelligence, for understanding how the neuromechanical control system mediates physical interaction to generate multi-pathway locomotor transitions in complex 3-D terrain. Note that our locomotor-terrain interaction system differs from protein folding in that animals and robots are macroscopic, self-propelled, far-from-equilibrium and can have intelligence. Image credits: (*a*) Left: from [83]. Reprinted with permission from AAAS. Right: adapted with permission from [84]. Copyright © (2012) American Chemical Society. (*b*) Right: copyright © IOP Publishing. Reproduced with permission from [4]. All rights reserved.

complex three-dimensional terrains can be understood by composing larger-scale, higher-dimensional potential energy landscapes (figure 5) from simple landscapes of abstracted challenges (e.g. figure 1*d*–*f*). Our terrain treadmill experiments (figure 3*e*) are beginning to shed light on this [49]. Progress towards such an understanding will lead to advancement for several fields.

## (a) Envisioned advancement for physics

The empirically discovered physical principles of locomotor transitions using feed-forward self-propulsion (§4d) are surprisingly similar to those of microscopic multi-pathway protein folding transitions (see detail in [50]), where predictive free energy landscape theories have been successful [67–69]. This was unexpected, given the differences in scale and nature of the interaction (macroscopic contact forces in locomotion versus ionic and dipole interactions, hydrogen bonds, van der Waals forces, hydrophobic interactions in protein folding) [68].

We envision the creation of analogous potential energy landscape theories, but with the addition of intelligence (e.g. §4e,f), to understand and predict how the animal's nervous system or robot's sensing, control and planning systems mediate physical interaction to generate multi-pathway locomotor transitions (such as observed in [4]). The next step towards this is to model conservative forces using potential energy landscape gradients, add stochastic, non-conservative propulsive and dissipative forces that perturb the system to 'diffuse' across landscape barriers (analogous to [85], but with closed-loop control of the landscape over locomotor degrees of freedom), and simulate multi-pathway locomotor transitions. Systematic studies to understand the principles

of force sensing [51] will inform how to steer the system and modify the landscape to modulate transitions intelligently using sensory feedback control. Such new theories will help expand the physics of living systems to the organismal level and expand statistical physics to macroscopic, far-from-equilibrium, self-propelled (active) systems [65,66].

## (b) Envisioned advancement for dynamical systems theory

Our potential energy landscape approach provided a new conceptual way of thinking about locomotor modes beyond near-steady-state, limit cycle-like behaviour (e.g. walk, run and climb [5–7]) (electronic supplementary material, figure S8*a*). Locomotion in irregular terrain with repeated perturbations requires an animal to continually modify its behaviour, which cannot be described by limit cycles [61]. Our work demonstrated that, in the potential energy landscape-dominated regime, the system must destabilize from an attractive landscape basin to transition from one mode to another, and locomotor modes can be metastable [86], far-from-steady manoeuvers (e.g. electronic supplementary material, figure S8*c*). We foresee the creation of new dynamical systems theories of terrestrial locomotion [22] that are composed of multi-pathway transitions across modes attracted to both landscape basins attractors and limit cycles [87] (electronic supplementary material, figure S8*d*).

In addition, such new dynamical systems theories modelling physical interaction may be combined with those that model related processes and factors such as proprioception [88], external sensory cues (e.g. predators, prey, resources) [14,89], internal needs (e.g. hunger, mating) [90] and safety–risk tradeoffs [91]. This integration will elucidate how these

factors interplay with physical interaction to modulate animals' locomotor transition behaviour in complex environments.

## (c) Envisioned advancement for biology

Our potential energy landscape approach provides a means towards the first principle, physical understanding of the organization of locomotor behaviour, filling a critical knowledge gap. The field of movement ecology [14] makes field observations of trajectories of animals—often as a point mass (e.g. [92])—moving and making behavioural transitions in natural environments, because physical interactions are difficult to measure at such large scales. Recent progress in quantitative ethology advanced understanding of the organization of behaviours, often by quantifying kinematics in homogeneous, near-featureless laboratory environments (see [12,13] for reviews). Our work highlights the importance and feasibility of, and opens new avenues for, studying how stereotypy and organization of behaviour are constrained by an animal's direct physical interaction with realistic environments. Analysing the disconnectivity [69] of basins of future composed landscapes for multi-pathway transitions will reveal the hierarchy ('treeness' [93]) of locomotor modes.

In addition, there are opportunities to explore how physical interaction during locomotion impacts large-scale processes like predator–prey pursuit and migration where locomotor performance is crucial [94]. If future potential energy landscape theories can predict how locomotor performance depends on relevant system parameters (§4d–g), they will provide a proxy for fitness landscapes [95]. Such proxy fitness landscapes will reveal how locomotor fitness exerts selective pressure on morphology and behaviour that affect locomotor transitions via physical interaction.

## (d) Envisioned advancement for robotics

Future predictive potential energy landscape theories will predict strategies for robots to use physical interaction to generate landscape basin attractors funnelled into one another [96] to compose locomotor transitions to perform high-level, goal-directed tasks in the real world. Using information of the geometry and physical properties of complex three-dimensional terrain from sensors, a robot can abstract its locomotor task into separate locomotor challenges (figure 1e) and calculate their potential energy landscapes. Then, the robot can use the landscape theories to identify possible transitions (figure 1d) and predict how transition probabilities differ between strategies (figure 4a,b). Finally, within its own constraints (e.g. energy available and actuator force limits), the robot can plan its strategies to make transitions that increase or even optimize its probability to reach the goal (figure 1d). When the terrain is sensed only up to a finite horizon with uncertainty, the robot can react to newly sensed challenges or recently failed attempts and update the pre-planned locomotor transition sequence and strategies (analogous to reactive obstacle avoidance using geometry [97]).

Recent learning approaches have managed to generate slow locomotion where terrain perturbations are sufficiently small for the learned controller to reject and stabilize the robot around an upright body posture [98,99]. Although learning approaches can in principle train the robot for any task in simulation by brute force, even in such modest terrain, the real system's physics must still be modelled properly (e.g. how motor dynamics affects leg dynamics) to narrow the simulation-to-reality gap [98,99]. However, as our work reveals, a robot should use physical interaction to destabilize itself to make locomotor transitions to traverse large obstacles. In addition, locomotor transitions are diverse and stochastic, and they depend sensitively on locomotor and terrain parameters and vary substantially with strategies. Considering these, learning approaches alone will be fragile for generating robot locomotor transitions in complex three-dimensional terrain. Our physics approach will be crucial for applying learning approaches here—it not only enables robots with basic transition capabilities (§4f; electronic supplementary material, table S1) to serve as real platforms for learning, but also offers principles of how strategies affect transitions across the large locomotor and terrain parameter space (§4g) to guide learning.

In the longer term, we envision that first principle models of locomotor–terrain physical interaction will be pervasive. Analogous to self-driving cars that scan streets, robots will create environmental physics maps and action databases for locomotor transitions and add them to geometric maps in the cloud for shared use [100]. They will help robots better use physical interaction to traverse currently unreachable complex three-dimensional terrain and expand our reaches in natural, artificial and extraterrestrial terrain.

Data accessibility. Data and code are available at https://github.com/TerradynamicsLab/potential_energy_landscape. An overview video is available at https://doi.org/10.6084/m9.figshare.14207927.

Authors' contributions. R.O. created visualizations and wrote the paper, Q.X. and Y.W. provided feedback, and C.L. revised the paper.

Competing interests. The authors declare no competing interests.

Funding. This work was funded by an Army Research Office Young Investigator Program (grant no. W911NF-17-1-0346), a Burroughs Wellcome Fund Career Award at the Scientific Interface, an Arnold and Mabel Beckman Foundation Beckman Young Investigator award and The Johns Hopkins University Whiting School of Engineering start-up funds to C.L.

Acknowledgements. We thank previous members of the Terradynamics lab, especially Sean Gart and Yuanfeng Han, for contribution to the studies reviewed here, Dan Goldman, Simon Sponberg, Bob Full, Dan Koditschek, Shai Revzen and Noah Cowan for discussion, and three anonymous reviewers for suggestions.

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
