## [Peer Review File · Proceedings of the Royal Society B: Biological Sciences]

Review History

RSPB-2020-2734.R0 (Original submission)

Review form: Reviewer 1

Recommendation

Accept with minor revision (please list in comments)

Scientific importance: Is the manuscript an original and important contribution to its field?

Good

General interest: Is the paper of sufficient general interest?

Good

Quality of the paper: Is the overall quality of the paper suitable?

Excellent

Is the length of the paper justified?

Yes

Should the paper be seen by a specialist statistical reviewer?

No

Do you have any concerns about statistical analyses in this paper? If so, please specify them explicitly in your report.

No

It is a condition of publication that authors make their supporting data, code and materials available - either as supplementary material or hosted in an external repository. Please rate, if applicable, the supporting data on the following criteria.

Is it accessible?

N/A

Is it clear?

N/A

Is it adequate?

N/A

Do you have any ethical concerns with this paper?

No

Comments to the Author

This paper reviews the authors' recent work on studying transitional dynamics of locomotor-terrain interactions. Specifically, the paper discusses a potential energy landscape modelling approach that allows computation of the system's potential energy in the relevant parameter space. Using case studies from their recent publications such as large obstacle traversal and self righting, the authors demonstrate that given the relevant parameter space identified through experiments, the potential energy landscape model can help identify different pathways that allows the system to passively descent towards lower-energy state, and suggest locomotor strategies that can help the system actively overcome potential energy barriers to increase the probability of falling into desired basins. The paper then provided a vision of how this approach can advance physics, biology, and robotics going forward.

It is an interesting article. Transitional locomotor-terrain interaction dynamics far away from steady state is not yet well studied, but is indeed of great importance to the locomotion science community. This article will be of relevance and interest to many researchers in the field.

The paper is well written. The visuals are clear and intuitive to understand; case studies are well presented, and show good promises of the new approach. The authors also exhibited good knowledge on a broad range of literature, and did a great job summarizing them.

My only suggestion is to condense the listing of detailed experimental tools and setups (Sec. 2), and elaborate more on how these methods differs from existing ones, and why they enable better understanding of transitional locomotion dynamics that was challenging before. This way, the readers can benefit more from the authors' in-depth perspectives and interpretations that were not directly available or clear from the reviewed work themselves.

Review form: Reviewer 2 (Shinya Aoi)

Recommendation

Accept with minor revision (please list in comments)

Scientific importance: Is the manuscript an original and important contribution to its field?

Excellent

General interest: Is the paper of sufficient general interest?

Good

Quality of the paper: Is the overall quality of the paper suitable?

Good

Is the length of the paper justified?

Yes

Should the paper be seen by a specialist statistical reviewer?

No

Do you have any concerns about statistical analyses in this paper? If so, please specify them explicitly in your report.

No

It is a condition of publication that authors make their supporting data, code and materials available - either as supplementary material or hosted in an external repository. Please rate, if applicable, the supporting data on the following criteria.

Is it accessible?

No

Is it clear?

Yes

Is it adequate?

Yes

Do you have any ethical concerns with this paper?

No

Comments to the Author

This manuscript reviews the potential energy landscape approach proposed and investigated in multiple studies by the authors to understand the locomotion strategies of animals to traverse complicated obstacles and to provide design principles for robust locomotion of legged robots. While traversing complicated obstacles is crucial for locomotion, there is no general methodology to investigate it. The proposed energy landscape approach is simple, clear, and suggestive. This manuscript is easy to read and understand and worth for publication. But, the following comments should be considered before publication.

Details of the potential energy landscape (especially, how to calculate it and how to use it) seem unclear. While the definition and calculation method are explained in the supplementary information for the gap obstacle, they are referred only in the caption of Fig. 4.

The barriers and basins must be crucial to understand the locomotor transitions from the energy landscape. However, their detailed structures and formation process seem unclear in Fig. 4. Fig. S2c for the gap obstacle is easy to find them. The same figure for other obstacles would be helpful.

The obstacles are categorized into five types. For each obstacle type, physics studies, biological studies, and robotics studies are conducted. The integration of the results and discussion for each obstacle type by following the concept in Fig. 2 would be useful.

Other specific comments are shown below.

L55, Fig. 1d

It is unclear what the three icons (tiger, biohazard, and fire) indicate.

L55, Fig. 1d

It is unclear why the self-right appears between flimsy beams and short bump.

L88, Fig. 2

This figure has a lot of information, but there are few explanations. More detailed explanations associated with individual studies would be helpful.

L137

"figure 3d" must be "figure 3e".

L223

Figures for the models would be helpful.

L224

Which parts of Fig. 2 do the studies of templates correspond to?

L267, Fig. 4b

It is difficult to find potential energy barriers and basins of each gait.

L267, Fig. 4a, b

Self-righting has three arrows in 4a, but two arrows in 4b.

L267, Fig. 4

Each obstacle must have some parameters, such as the gap length and depth of the gap obstacle. The transition strategies must change depending on the parameters. Have you ever investigated them?

L268, Fig. S2a

"circle ii'" must be "circle ii".

L268, Fig. S2cii

There are two circle ii'. The upper circle ii' must be circle ii.

L268, Fig. S2a, c

Are i, ii, iii in Fig. S2a circles i, ii, iii identical to i, ii, iii of Fig. S2ci, ii, iii? Fig. S2a circle iii seems to have already cleared the gap, while Fig. S2cii seems to show the potential energy landscape before clearing the gap.

L268, Fig. S2ciii

While showing the barriers are helpful to find the region of each basin in comparison with Fig. 4b, it remains difficult to find the region.

L373, Fig. 5c

While normal locomotion can be assumed as a limit cycle, what are the pitch and roll attractors?

L373, Fig. 5c

Why there are two circle 3?

L398

At the present time, the potential energy landscapes are described by two parameters and are

successfully visualized. Is this approach applicable when the number of required parameters increases?

Review form: Reviewer 3

Recommendation

Major revision is needed (please make suggestions in comments)

Scientific importance: Is the manuscript an original and important contribution to its field?

Excellent

General interest: Is the paper of sufficient general interest?

Good

Quality of the paper: Is the overall quality of the paper suitable?

Good

Is the length of the paper justified?

Yes

Should the paper be seen by a specialist statistical reviewer?

No

Do you have any concerns about statistical analyses in this paper? If so, please specify them explicitly in your report.

No

It is a condition of publication that authors make their supporting data, code and materials available - either as supplementary material or hosted in an external repository. Please rate, if applicable, the supporting data on the following criteria.

Is it accessible?

N/A

Is it clear?

N/A

Is it adequate?

N/A

Do you have any ethical concerns with this paper?

No

Comments to the Author

This review provides a synthesis of a relatively new concept in understanding locomotion in biology: the use of potential energy landscapes to understand transitions between modes. I don't know if the PI originated such analyses, but his work has certainly brought them to light. I think the lines that best summarize what this work is about are:

“However, there remains a knowledge gap in how locomotor transitions in complex terrain emerge from direct physical interaction (i.e., terradynamics [23]) of an animal's body and appendages with the environment controlled by the nervous system.”

and

“Because our potential energy landscapes are directly derived from first principles (as opposed to fitting a model to behavioral data [96–98]), our discovery that rugged potential energy landscapes result in stereotyped locomotor modes and transitions provided compelling evidence that behavioral stereotypy of animals emerges from their neural and mechanical systems directly interacting with the physical environment [14,15].”

Overall, this manuscript represents the cutting edge of the integration of mechanics and biology to understand how animals move on land. The review represents an in-depth summary of where the field is, helping to point it in new directions. I think it will be a very useful contribution to the literature.

Despite my enthusiasm, I have a large number of comments to address, both general and specific.

General comments:

From line 188: “However, our main modeling approach of potential energy landscape modeling is directly inspired by microscopic, multi-pathway protein folding transitions [79–81] (figure S1).” This approach is also reminiscent of evolutionary analyses of adaptive landscapes. Is this analogy not appropriate? Some others are mentioned later on and are described as not being good analogies, but it’s not really clear to me why that is. Some explanation is warranted.

Many of the analyses and discussion in this manuscript assume that the locomotor landscape is horizontal, but I don’t remember if that qualification is made anywhere.

The word ‘surprisingly’ is used in many places in the manuscript (e.g., line 24). For the most part, I didn’t understand what was surprising in these situations. If that word is to be retained, some explanation of why that thing is surprising is warranted.

The supplementary movie is interesting, but it is quite hard to follow because so many things happen at once. Perhaps break up some of the movement in some way, either temporally or by isolating spatially.

Specific comments by line number:

25 “General principles of locomotor transitions begin to emerge for this potential energy landscape dominated regime”

Can you provide some of the major examples, even briefly?

36 “Despite this, far more...”

What does ‘this’ refer to? Please specify to reduce ambiguity. This construction is used in multiple places in the manuscript.

39 “Previous studies have begun to reveal how terrestrial animals stochastically transition across locomotor modes in complex environments.”

How does an animal transition in a stochastic manner? Please explain.

57 Traversing complex 3D what?

84 The references should refer to ‘design and control strategies’, not design and control.

86 “Inspired by these recent successes, our group has been creating terradynamics of

locomotor transitions in complex 3-D terrain”

This usage is puzzling – how exactly has the group been creating terradynamics?

119 Change to “in fluids) for each model”.

137 “To study these, we developed a novel terrain treadmill with modular large obstacles (figure 3d) which records the animal motion at high spatial resolution [64].”

The treadmill records at high spatial resolution?

164 “In addition, terrain interaction of robotic physical models are governed by real physics (i.e., “you get the physics for free” [68]).”

The meaning of the quote in parentheses is not obvious – please explain what it means.

179 “Locomotor transitions in obstacle traversal and self-righting occur probabilistically via multiple pathways [58].”

Similar to the previous comment, what does it mean that they occur probabilistically? It’s not explained in this paragraph.

192 Change leads to lead.

223 “First, we developed dynamic templates”

I believe this phrase should be ‘dynamical templates’ or ‘templates of dynamics’.

231 “which landscapes cannot yet do due to the lack of system dynamics.”
Do you mean due to the lack of systems dynamical analyses?

233 “In addition, we developed multibody dynamics simulations of robotic physical model [63] to test the effect of randomness in wing-leg phase, which was not possible in the robotic model enabled larger- scale and finer variation of relevant parameters identified from animal and/or robot experiments for more in-depth measurements and dynamic analysis and revealed nuanced effects that may not be testable in robotic physical models like controlled variation of randomness.”

I don’t understand this long sentence.

250 Change has to have.

256 “which breaks continuous frictional contact and makes the system statically unstable.”
Is this always true? For example, what about insects that always have 3 legs in contact with the ground, or turtles when walking?

285 “However, outside of this regime, energy landscape modeling is not useful – for example, not for ballistic jumping over small obstacles with kinetic energy far exceeding potential energy barriers.”

Interesting point, but tell us more – what are other real-world specific examples?

308 Change provide to provides.

314 Change mode to modes.

315 Change to robots to go into.

325 Change constrains to constrain.

329 “We believe that this physical constraint not only has played a role in the evolution of animal morphology, physiology, and behavior for large obstacle traversal”

Do you have evidence of this statement of belief, or is it meant to be posed as a hypothesis?

391 “New energy landscape theories of multi-pathway locomotor transitions will help expand statistical physics to macroscopic, far-from-equilibrium, self-propelled (active) systems [76–78] and help establish physics of living systems at the organismal level.”

What does ‘establish’ mean here? The implication is that physics of living systems at the organismal level is not a recognized area. I would disagree with that contention. For one, there is a recognized NSF program of the same name that has funded organismal projects for years.

429 “It can then plan its strategies in one- shot”

I don’t know what this means.

445 Add a period after the citation.

483 Capitalize the book title, here and in other references.

569 Italicize the species name.

532 As a journal article, the title should not be capitalized, here and in other references.

575, 577 Correct the numerical information.

603 Is this reference in review or in press?

696 Capitalize and italicize the genus name, here and in other references.

Supplement

28 A ‘bridge’ implies that the animals body fully subtends the gap, but animals cross gaps in manners relevant to this manuscript in ways that go beyond bridging. ‘Gap crossing’ is a more general and appropriate term. See Graham 2020. *J. Exp. Zool. A*: 333(1):60-73.

Decision letter (RSPB-2020-2734.R0)

03-Dec-2020

Dear Mr Othayoth:

Your manuscript has now been peer reviewed and the reviews have been assessed by an Associate Editor. The reviewers’ comments (not including confidential comments to the Editor) and the comments from the Associate Editor are included at the end of this email for your reference. As you will see, the reviewers and the Editors have raised some concerns with your manuscript and we would like to invite you to revise your manuscript to address them.

We do not allow multiple rounds of revision so we urge you to make every effort to fully address all of the comments at this stage. If deemed necessary by the Associate Editor, your manuscript will be sent back to one or more of the original reviewers for assessment. If the original reviewers

are not available we may invite new reviewers. Please note that we cannot guarantee eventual acceptance of your manuscript at this stage.

Research ethics:

Use of animals and field studies:

It is a condition of publication that you make available the data and research materials supporting the results in the article. Please see our Data Sharing Policies (<https://royalsociety.org/journals/authors/author-guidelines/#data>). Datasets should be deposited in an appropriate publicly available repository and details of the associated accession number, link or DOI to the datasets must be included in the Data Accessibility section of the article (<https://royalsociety.org/journals/ethics-policies/data-sharing-mining/>). Reference(s) to datasets should also be included in the reference list of the article with DOIs (where available).

Please submit a copy of your revised paper within three weeks. If we do not hear from you within this time your manuscript will be rejected. If you are unable to meet this deadline please let us know as soon as possible, as we may be able to grant a short extension.

Best wishes,

Dr Locke Rowe

Associate Editor

Board Member: 1

Comments to Author:

The reviewers and guest editor agree that this is an interesting and generally well-written article that will be a great addition to the special issue. The reviewers provide good comments and suggestions for improvements and clarifications. Please address those points carefully. It will improve the readability and impact of the article.

Reviewer(s)' Comments to Author:

Referee: 1

Comments to the Author(s)

This paper reviews the authors' recent work on studying transitional dynamics of locomotor-terrain interactions. Specifically, the paper discusses a potential energy landscape modelling approach that allows computation of the system's potential energy in the relevant parameter space. Using case studies from their recent publications such as large obstacle traversal and self righting, the authors demonstrate that given the relevant parameter space identified through experiments, the potential energy landscape model can help identify different pathways that allows the system to passively descent towards lower-energy state, and suggest locomotor strategies that can help the system actively overcome potential energy barriers to increase the probability of falling into desired basins. The paper then provided a vision of how this approach can advance physics, biology, and robotics going forward.

It is an interesting article. Transitional locomotor-terrain interaction dynamics far away from steady state is not yet well studied, but is indeed of great importance to the locomotion science community. This article will be of relevance and interest to many researchers in the field.

The paper is well written. The visuals are clear and intuitive to understand; case studies are well presented, and show good promises of the new approach. The authors also exhibited good knowledge on a broad range of literature, and did a great job summarizing them.

My only suggestion is to condense the listing of detailed experimental tools and setups (Sec. 2), and elaborate more on how these methods differs from existing ones, and why they enable better

understanding of transitional locomotion dynamics that was challenging before. This way, the readers can benefit more from the authors' in-depth perspectives and interpretations that were not directly available or clear from the reviewed work themselves.

Referee: 2

Comments to the Author(s)

This manuscript reviews the potential energy landscape approach proposed and investigated in multiple studies by the authors to understand the locomotion strategies of animals to traverse complicated obstacles and to provide design principles for robust locomotion of legged robots. While traversing complicated obstacles is crucial for locomotion, there is no general methodology to investigate it. The proposed energy landscape approach is simple, clear, and suggestive. This manuscript is easy to read and understand and worth for publication. But, the following comments should be considered before publication.

Details of the potential energy landscape (especially, how to calculate it and how to use it) seem unclear. While the definition and calculation method are explained in the supplementary information for the gap obstacle, they are referred only in the caption of Fig. 4.

The barriers and basins must be crucial to understand the locomotor transitions from the energy landscape. However, their detailed structures and formation process seem unclear in Fig. 4. Fig. S2c for the gap obstacle is easy to find them. The same figure for other obstacles would be helpful.

The obstacles are categorized into five types. For each obstacle type, physics studies, biological studies, and robotics studies are conducted. The integration of the results and discussion for each obstacle type by following the concept in Fig. 2 would be useful.

Other specific comments are shown below.

L55, Fig. 1d

It is unclear what the three icons (tiger, biohazard, and fire) indicate.

L55, Fig. 1d

It is unclear why the self-right appears between flimsy beams and short bump.

L88, Fig. 2

This figure has a lot of information, but there are few explanations. More detailed explanations associated with individual studies would be helpful.

L137

"figure 3d" must be "figure 3e".

L223

Figures for the models would be helpful.

L224

Which parts of Fig. 2 do the studies of templates correspond to?

L267, Fig. 4b

It is difficult to find potential energy barriers and basins of each gait.

L267, Fig. 4a, b

Self-righting has three arrows in 4a, but two arrows in 4b.

L267, Fig. 4

Each obstacle must have some parameters, such as the gap length and depth of the gap obstacle. The transition strategies must change depending on the parameters. Have you ever investigated them?

L268, Fig. S2a

"circle ii'" must be "circle ii".

L268, Fig. S2cii

There are two circle ii'. The upper circle ii' must be circle ii.

L268, Fig. S2a, c

Are i, ii, iii in Fig. S2a circles i, ii, iii identical to i, ii, iii of Fig. S2ci, ii, iii? Fig. S2a circle iii seems to have already cleared the gap, while Fig. S2cii seems to show the potential energy landscape before clearing the gap.

L268, Fig. S2ciii

While showing the barriers are helpful to find the region of each basin in comparison with Fig. 4b, it remains difficult to find the region.

L373, Fig. 5c

While normal locomotion can be assumed as a limit cycle, what are the pitch and roll attractors?

L373, Fig. 5c

Why there are two circle 3?

L398

At the present time, the potential energy landscapes are described by two parameters and are successfully visualized. Is this approach applicable when the number of required parameters increases?

Referee: 3

Comments to the Author(s)

This review provides a synthesis of a relatively new concept in understanding locomotion in biology: the use of potential energy landscapes to understand transitions between modes. I don't know if the PI originated such analyses, but his work has certainly brought them to light. I think the lines that best summarize what this work is about are:

"However, there remains a knowledge gap in how locomotor transitions in complex terrain emerge from direct physical interaction (i.e., terradynamics [23]) of an animal's body and appendages with the environment controlled by the nervous system."

and

"Because our potential energy landscapes are directly derived from first principles (as opposed to fitting a model to behavioral data [96-98]), our discovery that rugged potential energy landscapes result in stereotyped locomotor modes and transitions provided compelling evidence that behavioral stereotypy of animals emerges from their neural and mechanical systems directly interacting with the physical environment [14,15]."

Overall, this manuscript represents the cutting edge of the integration of mechanics and biology to understand how animals move on land. The review represents an in-depth summary of where the field is, helping to point it in new directions. I think it will be a very useful contribution to the literature.

Despite my enthusiasm, I have a large number of comments to address, both general and specific.

General comments:

From line 188: “However, our main modeling approach of potential energy landscape modeling is directly inspired by microscopic, multi-pathway protein folding transitions [79–81] (figure S1).”

This approach is also reminiscent of evolutionary analyses of adaptive landscapes. Is this analogy not appropriate? Some others are mentioned later on and are described as not being good analogies, but it’s not really clear to me why that is. Some explanation is warranted. Many of the analyses and discussion in this manuscript assume that the locomotor landscape is horizontal, but I don’t remember if that qualification is made anywhere.

The word ‘surprisingly’ is used in many places in the manuscript (e.g., line 24). For the most part, I didn’t understand what was surprising in these situations. If that word is to be retained, some explanation of why that thing is surprising is warranted.

The supplementary movie is interesting, but it is quite hard to follow because so many things happen at once. Perhaps break up some of the movement in some way, either temporally or by isolating spatially.

Specific comments by line number:

25 “General principles of locomotor transitions begin to emerge for this potential energy landscape dominated regime”

Can you provide some of the major examples, even briefly?

36 “Despite this, far more...”

What does ‘this’ refer to? Please specify to reduce ambiguity. This construction is used in multiple places in the manuscript.

39 “Previous studies have begun to reveal how terrestrial animals stochastically transition across locomotor modes in complex environments.”

How does an animal transition in a stochastic manner? Please explain.

57 Traversing complex 3D what?

84 The references should refer to ‘design and control strategies’, not design and control.

86 “Inspired by these recent successes, our group has been creating terradynamics of locomotor transitions in complex 3-D terrain”

This usage is puzzling – how exactly has the group been creating terradynamics?

119 Change to “in fluids) for each model”.

137 “To study these, we developed a novel terrain treadmill with modular large obstacles (figure 3d) which records the animal motion at high spatial resolution [64].”

The treadmill records at high spatial resolution?

164 “In addition, terrain interaction of robotic physical models are governed by real physics (i.e., “you get the physics for free” [68]).”

The meaning of the quote in parentheses is not obvious – please explain what it means.

179 “Locomotor transitions in obstacle traversal and self-righting occur probabilistically via multiple pathways [58].”

Similar to the previous comment, what does it mean that they occur probabilistically? It’s not explained in this paragraph.

192 Change leads to lead.

223 “First, we developed dynamic templates”

I believe this phrase should be ‘dynamical templates’ or ‘templates of dynamics’.

231 “which landscapes cannot yet do due to the lack of system dynamics.”

Do you mean due to the lack of systems dynamical analyses?

233 “In addition, we developed multibody dynamics simulations of robotic physical model [63] to test the effect of randomness in wing-leg phase, which was not possible in the robotic model enabled larger- scale and finer variation of relevant parameters identified from animal and/or robot experiments for more in-depth measurements and dynamic analysis and revealed nuanced effects that may not be testable in robotic physical models like controlled variation of randomness.”

I don’t understand this long sentence.

250 Change has to have.

256 “which breaks continuous frictional contact and makes the system statically unstable.”

Is this always true? For example, what about insects that always have 3 legs in contact with the ground, or turtles when walking?

285 “However, outside of this regime, energy landscape modeling is not useful – for example, not for ballistic jumping over small obstacles with kinetic energy far exceeding potential energy barriers.”

Interesting point, but tell us more – what are other real-world specific examples?

308 Change provide to provides.

314 Change mode to modes.

315 Change to robots to go into.

325 Change constrains to constrain.

329 “We believe that this physical constraint not only has played a role in the evolution of animal morphology, physiology, and behavior for large obstacle traversal”

Do you have evidence of this statement of belief, or is it meant to be posed as a hypothesis?

391 “New energy landscape theories of multi-pathway locomotor transitions will help expand statistical physics to macroscopic, far-from-equilibrium, self-propelled (active) systems [76-78] and help establish physics of living systems at the organismal level.”

What does 'establish' mean here? The implication is that physics of living systems at the organismal level is not a recognized area. I would disagree with that contention. For one, there is a recognized NSF program of the same name that has funded organismal projects for years.

429 "It can then plan its strategies in one- shot"

I don't know what this means.

445 Add a period after the citation.

483 Capitalize the book title, here and in other references.

569 Italicize the species name.

532 As a journal article, the title should not be capitalized, here and in other references.

575, 577 Correct the numerical information.

603 Is this reference in review or in press?

696 Capitalize and italicize the genus name, here and in other references.

Supplement

28 A 'bridge' implies that the animals body fully subtends the gap, but animals cross gaps in manners relevant to this manuscript in ways that go beyond bridging. 'Gap crossing' is a more general and appropriate term. See Graham 2020. J. Exp. Zool. A: 333(1):60-73.

Author's Response to Decision Letter for (RSPB-2020-2734.R0)

See Appendix A.

RSPB-2020-2734.R1 (Revision)

Review form: Reviewer 1

Recommendation

Accept as is

Scientific importance: Is the manuscript an original and important contribution to its field?

Excellent

General interest: Is the paper of sufficient general interest?

Good

Quality of the paper: Is the overall quality of the paper suitable?

Excellent

Is the length of the paper justified?

Yes

Should the paper be seen by a specialist statistical reviewer?

No

Do you have any concerns about statistical analyses in this paper? If so, please specify them explicitly in your report.

No

It is a condition of publication that authors make their supporting data, code and materials available - either as supplementary material or hosted in an external repository. Please rate, if applicable, the supporting data on the following criteria.

Is it accessible?

N/A

Is it clear?

N/A

Is it adequate?

N/A

Do you have any ethical concerns with this paper?

No

Comments to the Author

The authors did a good job addressing the comments. I recommend this paper to be accepted.

Review form: Reviewer 2 (Shinya Aoi)

Recommendation

Accept as is

Scientific importance: Is the manuscript an original and important contribution to its field?

Excellent

General interest: Is the paper of sufficient general interest?

Excellent

Quality of the paper: Is the overall quality of the paper suitable?

Excellent

Is the length of the paper justified?

Yes

Should the paper be seen by a specialist statistical reviewer?

No

Do you have any concerns about statistical analyses in this paper? If so, please specify them explicitly in your report.

No

It is a condition of publication that authors make their supporting data, code and materials available - either as supplementary material or hosted in an external repository. Please rate, if applicable, the supporting data on the following criteria.

Is it accessible?

Yes

Is it clear?

Yes

Is it adequate?

Yes

Do you have any ethical concerns with this paper?

No

Comments to the Author

The manuscript is well revised based on the reviewers comments.

Review form: Reviewer 3 (Jake Socha)

Recommendation

Accept with minor revision (please list in comments)

Scientific importance: Is the manuscript an original and important contribution to its field?

Excellent

General interest: Is the paper of sufficient general interest?

Good

Quality of the paper: Is the overall quality of the paper suitable?

Good

Is the length of the paper justified?

Yes

Should the paper be seen by a specialist statistical reviewer?

No

Do you have any concerns about statistical analyses in this paper? If so, please specify them explicitly in your report.

No

It is a condition of publication that authors make their supporting data, code and materials available - either as supplementary material or hosted in an external repository. Please rate, if applicable, the supporting data on the following criteria.

Is it accessible?

Yes

Is it clear?

Yes

Is it adequate?

Yes

Do you have any ethical concerns with this paper?

No

Comments to the Author

The authors have done a good job of addressing the concerns of the reviewers. I don't have any further major comments. Below are a few remaining small considerations and suggestions that may help to improve the manuscript prior to publication.

Comment from the first submission:

The word 'surprisingly' is used in many places in the manuscript (e.g., line 24). For the most part, I didn't understand what was surprising in these situations. If that word is to be retained, some explanation of why that thing is surprising is warranted.

Author response:

The physical principles for macroscopic, far-from-equilibrium animals and robots making locomotor transitions are similar to those of microscopic, near-equilibrium protein-folding transitions on free energy landscapes. This was unexpected, given the differences in scale and nature of interaction (mechanical interaction in locomotion vs. van der Waals, dipole, hydrogen bond, hydrophobic, etc. in protein folding).

We removed all uses except the following instance:

Lines: 327-329 "Our empirically discovered physical principles of locomotor transitions are surprisingly similar to those of microscopic multi-pathway protein folding transitions where predictive energy landscape theories have been successful [63–65] (see detail in [44])."

Now that I have a better understanding of the authors' perspective, I still think this instance needs additional text. It would be much stronger if the reasons for finding it surprising are provided, rather than stating it baldly. Simply add the explanation given in the response above, and then it makes sense.

Comment from first submission:

This usage is puzzling – how exactly has the group been creating terradynamics?

Author response:

Terradynamics is the study of physical interaction during locomotion in complex terrain (Li, Zhang, Goldman, 2013 Science; Li, Pullin, Haldane, Lam, Fearing, Full, 2015, Bioinspiration & Biomimetics). We integrate animal and robot experiments to empirically observe the interaction and resulting locomotor transitions in complex 3-D terrain. We then create physics models such as the potential energy landscapes and dynamical templates to understand how locomotor transitions emerge from physical interaction. These are the subject of this review and elaborated throughout the paper.

I think that the authors misunderstood my comment. I wasn't looking for an explanation of terradynamics. As stated in this response, terradynamics is an area of inquiry. The current sentence in line 65 is "A physics-based approach by creating terradynamics [20] holds promise for filling this major gap." This sentence doesn't make sense to me. Substitute in other areas of inquiry, say "ecology" or "functional morphology" or "thermodynamics", and it similarly doesn't make sense. "Create" is the wrong word; I think "using" was intended.

Other suggestions by line:

- 20 Change to “understanding have advanced”.
- 29 What does “this” refer to?
- 33 Change to “animal behaviors”
- 59 Change to “near-flat”
- 69 Change to “dynamics techniques”
- 108 What does “these” refer to?
- 116 Change to “allows us” or “allows one”
- 271 What does “this” refer to?
- 275 Change to “landscape”
- 310 Change to “and do not”
- 357 What is “it”? What does “This” refer to?
- 376 Pull out the long parenthetical and make it into a sentence that follows, as in “For instance, dense forests...”

Decision letter (RSPB-2020-2734.R1)

08-Mar-2021

Dear Mr Othayoth

I am pleased to inform you that your manuscript RSPB-2020-2734.R1 entitled "Locomotor transitions in the potential energy landscape dominated regime" has been accepted for publication in Proceedings B.

The referee(s) have recommended publication, but also suggest some minor revisions to your manuscript. Therefore, I invite you to respond to the referee(s)' comments and revise your manuscript. Because the schedule for publication is very tight, it is a condition of publication that you submit the revised version of your manuscript within 7 days. If you do not think you will be able to meet this date please let us know.

When submitting your revised manuscript, you will be able to respond to the comments made by the referee(s) and upload a file "Response to Referees". You can use this to document any changes you make to the original manuscript. We require a copy of the manuscript with revisions made

since the previous version marked as 'tracked changes' to be included in the 'response to referees' document.

Sincerely,
Dr Locke Rowe
Editor, Proceedings B
<mailto:proceedingsb@royalsociety.org>

Associate Editor:

Board Member: 1

Comments to Author:

The article has been nicely revised. Well done. One reviewer still has some small and useful suggestions to improve readability. Please address those for the camera-ready version.

Reviewer(s)' Comments to Author:

Referee: 1

Comments to the Author(s)

The authors did a good job addressing the comments. I recommend this paper to be accepted.

Referee: 2

Comments to the Author(s)

The manuscript is well revised based on the reviewers comments.

Referee: 3

Comments to the Author(s)

The authors have done a good job of addressing the concerns of the reviewers. I don't have any further major comments. Below are a few remaining small considerations and suggestions that may help to improve the manuscript prior to publication.

Comment from the first submission:

The word 'surprisingly' is used in many places in the manuscript (e.g., line 24). For the most part, I didn't understand what was surprising in these situations. If that word is to be retained, some explanation of why that thing is surprising is warranted.

Author response:

The physical principles for macroscopic, far-from-equilibrium animals and robots making locomotor transitions are similar to those of microscopic, near-equilibrium protein-folding transitions on free energy landscapes. This was unexpected, given the differences in scale and nature of interaction (mechanical interaction in locomotion vs. van der Waals, dipole, hydrogen bond, hydrophobic, etc. in protein folding).

We removed all uses except the following instance:

Lines: 327-329 "Our empirically discovered physical principles of locomotor transitions are surprisingly similar to those of microscopic multi-pathway protein folding transitions where predictive energy landscape theories have been successful [63-65] (see detail in [44])."

Now that I have a better understanding of the authors' perspective, I still think this instance needs additional text. It would be much stronger if the reasons for finding it surprising are provided, rather than stating it baldly. Simply add the explanation given in the response above, and then it makes sense.

Comment from first submission:

This usage is puzzling – how exactly has the group been creating terradynamics?

Author response:

Terradynamics is the study of physical interaction during locomotion in complex terrain (Li, Zhang, Goldman, 2013 Science; Li, Pullin, Haldane, Lam, Fearing, Full, 2015, Bioinspiration & Biomimetics). We integrate animal and robot experiments to empirically observe the interaction and resulting locomotor transitions in complex 3-D terrain. We then create physics models such as the potential energy landscapes and dynamical templates to understand how locomotor transitions emerge from physical interaction. These are the subject of this review and elaborated throughout the paper.

I think that the authors misunderstood my comment. I wasn't looking for an explanation of terradynamics. As stated in this response, terradynamics is an area of inquiry. The current sentence in line 65 is "A physics-based approach by creating terradynamics [20] holds promise for filling this major gap." This sentence doesn't make sense to me. Substitute in other areas of inquiry, say "ecology" or "functional morphology" or "thermodynamics", and it similarly doesn't make sense. "Create" is the wrong word; I think "using" was intended.

Other suggestions by line:

20 Change to "understanding have advanced".

29 What does "this" refer to?

33 Change to "animal behaviors"

59 Change to "near-flat"

69 Change to "dynamics techniques"

108 What does "these" refer to?

116 Change to "allows us" or "allows one"

271 What does "this" refer to?

275 Change to "landscape"

310 Change to "and do not"

357 What is "it"? What does "This" refer to?

376 Pull out the long parenthetical and make it into a sentence that follows, as in "For instance, dense forests..."

Author's Response to Decision Letter for (RSPB-2020-2734.R1)

See Appendix B.

Decision letter (RSPB-2020-2734.R2)

26-Mar-2021

Dear Mr Othayoth

I am pleased to inform you that your manuscript entitled "Locomotor transitions in the potential energy landscape-dominated regime" has been accepted for publication in Proceedings B.

Data Accessibility section

Open Access

Paper charges

Sincerely,
Editor, Proceedings B
<mailto:proceedingsb@royalsociety.org>

Appendix A

We thank the reviewers for their comments.
Please see below for response to each of the points.

We also made small edits throughout the entire manuscript and substantially shortened it (at the request of journal staff) from 9959 words to 7516 words.

We believe the manuscript is substantially improved thanks to their suggestions.

Please note that the line numbers mentioned in the responses below are with respect to the document with tracked changes highlighted (attached at the end of this letter).

Associate Editor

Board Member: 1

Comments to Author:

The reviewers and guest editor agree that this is an interesting and generally well-written article that will be a great addition to the special issue. The reviewers provide good comments and suggestions for improvements and clarifications. Please address those points carefully. It will improve the readability and impact of the article.

Reviewer(s)' Comments to Author:

Referee: 1

Comments to the Author(s)

This paper reviews the authors' recent work on studying transitional dynamics of locomotor-terrain interactions. Specifically, the paper discusses a potential energy landscape modelling approach that allows computation of the system's potential energy in the relevant parameter space. Using case studies from their recent publications such as large obstacle traversal and self righting, the authors demonstrate that given the relevant parameter space identified through experiments, the potential energy landscape model can help identify different pathways that allows the system to passively descent towards lower-energy state, and suggest locomotor strategies that can help the system actively overcome potential energy barriers to increase the probability of falling into desired basins. The paper then provided a vision of how this approach can advance physics, biology, and robotics going forward.

It is an interesting article. Transitional locomotor-terrain interaction dynamics far away from steady state is not yet well studied, but is indeed of great importance to the locomotion science community. This article will be of relevance and interest to many researchers in the field.

The paper is well written. The visuals are clear and intuitive to understand; case studies are well presented, and show good promises of the new approach. The authors also exhibited good knowledge on a broad range of literature, and did a great job summarizing them.

My only suggestion is to condense the listing of detailed experimental tools and setups (Sec. 2), and elaborate more on how these methods differs from existing ones, and why they enable better understanding of transitional locomotion dynamics that was challenging before. This way, the readers can benefit more from the authors' in-depth perspectives and interpretations that were not directly available or clear from the reviewed work themselves.

We shortened and this section and moved most detail to Supplementary Information.

To clarify, we are not claiming that our techniques are brand new. However, they are a step forward compared to terrestrial locomotion studies that quantify 2-D or 3-D movement in relatively simple environment. We elaborated this in the revised paragraphs below by adding more detail to quantify the improvements.

Lines 96-118 in Supplementary Information:

“There are several technical challenges to measuring locomotor transitions and locomotor-terrain interaction during traversal of complex 3-D terrain and self-righting. First, the traditional technique of using two or three cameras for measuring 3-D motion on flat surfaces is inadequate to record the large range of 3-D rotations of the animal, robot, or terrain (if any) or cope with frequent occlusion, both common in complex 3-D terrain [5]. For example, the animal could be tracked in only 24%, 60%, or 77% of the frames during a beam traversal attempt with only 2, 3, or 4 cameras. In addition, manually measuring locomotor and terrain kinematics is laborious, especially for large datasets ($\sim 10^2$ - 10^3 trials) required for making statistically meaningful conclusions. For example, for the ~ 300 animal trials in our study of beam traversal [6], it would take 500 hours to manually track the animal body and two beams. Furthermore, to measure appendage motion, body-obstacle perturbations, and frequent collisions that are smaller than body/obstacle size, we need high accuracy 3-D kinematics. For example, a tracking error of ± 1 mm in the animal’s position during beam interaction can over or underestimate beam deflection angle by $\sim 13^\circ$.

We developed several tools to address these challenges. First, we developed an automated visible light imaging system (figure 3b) with up to 12 synchronized high-resolution, high-speed cameras [6,18]. This system can capture near-continuous locomotor-terrain interaction over the full range of 3-D translation and rotation and track the animal in more than 96% of the interaction phase. In addition, we reduced the manual labor for calibration and 3-D motion reconstruction by 250 times by automatically tracking custom calibration objects (figure 3c) and animal and terrain motions (figure 3d) in each camera view, using uniquely distinguishable QR-code markers (figure 3a) [19]. Furthermore, to ensure high accuracy measurement of terrain interactions, we designed custom calibration objects to span the entire field of view of all 12 cameras and used a high precision 3-D printed object to verify tracking and reconstruction fidelity (s.d. of position error = 0.6 mm; s.d. of orientation error = 1.1°).”

Referee: 2

Comments to the Author(s)

This manuscript reviews the potential energy landscape approach proposed and investigated in multiple studies by the authors to understand the locomotion strategies of animals to traverse complicated obstacles and to provide design principles for robust locomotion of legged robots. While traversing complicated obstacles is crucial for locomotion, there is no general methodology to investigate it. The proposed energy landscape approach is simple, clear, and suggestive. This manuscript is easy to read and understand and worth for publication. But, the following comments should be considered before publication.

Details of the potential energy landscape (especially, how to calculate it and how to use it) seem unclear. While the definition and calculation method are explained in the supplementary information for the gap obstacle, they are referred only in the caption of Fig. 4.

The barriers and basins must be crucial to understand the locomotor transitions from the energy landscape. However, their detailed structures and formation process seem unclear in Fig. 4. Fig. S2c for the gap obstacle is easy to find them. The same figure for other obstacles would be helpful.

We added general steps to calculate potential energy landscape and measuring system behavior.

See Supplementary Text S2 and Supplementary Figure S2.

We also added assumptions, calculations, and detailed schematic figures for energy landscapes of all model systems.

See Supplementary Text S3 to S7 and Supplementary Figures S3-S7.

The obstacles are categorized into five types. For each obstacle type, physics studies, biological studies, and robotics studies are conducted. The integration of the results and discussion for each obstacle type by following the concept in Fig. 2 would be useful.

Here in the review, we focus on the common features of energy landscapes (Section 4).

For each model system, we briefly summarized the major observations, physical principles discovered, and their implications for robotics in Supplementary Table S1.

Other specific comments are shown below.

L55, Fig. 1d

It is unclear what the three icons (tiger, biohazard, and fire) indicate.

Removed.

L55, Fig. 1d

It is unclear why the self-righting appears between flimsy beams and short bump.

In complex 3-D terrain, animals and robots are susceptible to flipping over due to large obstacle interaction (Li et al. (2017) *Advanced Robotics*). For example, in a previous study (Li et al. (2015) *Bioinspiration and Biomimetics*), legged robots traversing layers of beam obstacles sometimes flipped over when emerging from the gap between beams. So, we include it to illustrate this possibility and the need for this transition.

We revised the arrow in figure 1d to clarify that the body is not propelled forward significantly during self-righting, compared to other modes.

L88, Fig. 2

This figure has a lot of information, but there are few explanations. More detailed explanations associated with individual studies would be helpful.

We added figure caption to explain the approach.

We did not add detailed explanations for each study as we feel it would be excessive (there are a total of 12 studies and each one often involves several aspects shown in Fig. 2). However, we added Supplementary Table S1 as mentioned above.

L137

"figure 3d" must be "figure 3e".

Fixed.

Line 109.

L223

Figures for the models would be helpful.

Model figures and calculations have been added to Supplementary Materials.

L224

Which parts of Fig. 2 do the studies of templates correspond to?

Dynamical templates belong to qualitative predictive models. They can reveal mechanisms of animal locomotion and provide design tools and action strategies for robot locomotion.

L267, Fig. 4b

It is difficult to find potential energy barriers and basins of each gait.

Here due to the small size it is difficult to elaborate all these features.

We added detailed figures of landscape basins and barriers in Supplementary figures S3-S7.

L267, Fig. 4a, b

Self-righting has three arrows in 4a, but two arrows in 4b.

Fixed.

L267, Fig. 4

Each obstacle must have some parameters, such as the gap length and depth of the gap obstacle. The transition strategies must change depending on the parameters. Have you ever investigated them?

Yes, our previous studies have investigated how locomotor transitions and their relative frequency depends on bump height (Gart and Li (2018), *Bioinspiration and Biomimetics*), gap width (Gart et al (2018), *Bioinspiration and Biomimetics*), beam stiffness (Othayoth et al (2020), *PNAS*), pillar shape, size, and orientation (Han et al (in press), *IJRR*).

In this review, we focus on discussing common features from the energy landscape approach. However, we briefly summarized the major results from other studies in Supplementary Table S1.

L268, Fig. S2a

"circle ii'" must be "circle ii".

Fixed; see Figure S3.

L268, Fig. S2cii

There are two circle ii'. The upper circle ii' must be circle ii.

Fixed; see Figure S3.

L268, Fig. S2a, c

Are i, ii, iii in Fig. S2a circles i, ii, iii identical to i, ii, iii of Fig. S2ci, ii, iii? Fig. S2a circle iii seems to have already cleared the gap, while Fig. S2ciii seems to show the potential energy landscape before clearing the gap.

Yes. The positions of the ellipsoid that cleared the gap (Fig. S2aiii) and the one deflecting (Fig. S2aiii') have been corrected to be at the same forward positions (x) so that are represented on the same landscape snapshot in Fig. S2c.

Fixed; see Figure S3.

L268, Fig. S2ciii

While showing the barriers are helpful to find the region of each basin in comparison with Fig. 4b, it remains difficult to find the region.

We revised Fig. 4b to better clarify energy basins and trajectories corresponding to various modes and added energy barriers that separate these basins.

L373, Fig. 5c

While normal locomotion can be assumed as a limit cycle, what are the pitch and roll attractors?

Pitch and roll attractors shown in Fig. 5c are spiral sinks in a 2-D linear dynamical system. This is purely speculative, and the nature/type of attractive landscape basins remains to be studied. Revised Fig. 5 caption to clarify.

Lines 313-314:

“Spiral sinks [84] are used as a speculative schematic for attractive landscape basins in figure 4b, iv; their exact nature remains to be discovered”

L373, Fig. 5c

Why there are two circle 3?

The two circles were meant to be (3) and (3'), corresponding to beam traversal by pitch and roll modes, respectively. Revised the numbering to be consistent with other figures.

L398

At the present time, the potential energy landscapes are described by two parameters and are successfully visualized. Is this approach applicable when the number of required parameters increases?

When the number of required parameters increases, visualization of the landscape over the entire required parameter space becomes more difficult or even impossible.

Nevertheless, similar analysis is still possible using higher dimensional landscapes. This is commonly done in high-dimensional landscape analysis such as free energy landscapes of protein folding (Wales (2003), *Energy Landscapes: Applications to Clusters, Biomolecules and Glasses*).

For three parameters, the evolving landscape and its basins can be better visualized using a video, with one of the parameters changing as the video proceeds (e.g., Supplementary Video S1 shows how the landscape for beam obstacle traversal over body roll and pitch space further changes with increasing forward position of the body).

Alternatively, for three parameters, a contour can be chosen along any two dimensions, and its variation along the third dimension can be visualized as a surface. This technique has been used in adaptive landscapes (see Figure 5 in Philips and Arnold (1989), *Visualizing multivariate selection, Evolution*).

Lines 143-155 in Supplementary Materials:

4. “Choose the few system state degrees of freedom over which the energy landscape is to be constructed. Often degrees of freedom that represent self-propulsion (e.g., body forward position relative to obstacles, wing opening angle in ground self-righting) and those that change substantially in response to terrain interaction (e.g., body pitch, roll, yaw) are chosen. Because a high-dimensional landscape over a large number of degrees of freedom is more challenging to understand, three degrees of freedom can be chosen first to visualize the landscape more easily, as a potential energy map over two degrees of freedom which further evolves over the third degree of freedom (e.g., Supplementary Movie S1).
5. Construct the potential energy landscape over the first two chosen degrees of freedom while keeping the third (and other) degrees of freedom constant. Varying both these degrees of freedom in discrete increments over the desired range and calculating the potential energy at each point of the grid in this 2-D parameter space.
6. Construct the evolving potential energy landscape by repeating steps 5 while varying the third (and remaining) degrees of freedom, either along a prescribed or experimentally measured trajectory.”

Referee: 3

Comments to the Author(s)

This review provides a synthesis of a relatively new concept in understanding locomotion in biology: the use of potential energy landscapes to understand transitions between modes. I don't know if the PI originated such analyses, but his work has certainly brought them to light. I think the lines that best summarize what this work is about are:

“However, there remains a knowledge gap in how locomotor transitions in complex terrain emerge from direct physical interaction (i.e., terradynamics [23]) of an animal's body and appendages with the environment controlled by the nervous system.”

and

“Because our potential energy landscapes are directly derived from first principles (as opposed to fitting a model to behavioral data [96–98]), our discovery that rugged potential energy landscapes result in stereotyped locomotor modes and transitions provided compelling evidence that behavioral stereotypy of animals emerges from their neural and mechanical systems directly interacting with the physical environment [14,15].”

Overall, this manuscript represents the cutting edge of the integration of mechanics and biology to understand how animals move on land. The review represents an in-depth summary of where the field is, helping to point it in new directions. I think it will be a very useful contribution to the literature.

Despite my enthusiasm, I have a large number of comments to address, both general and specific.

General comments:

From line 188: “However, our main modeling approach of potential energy landscape modeling is directly inspired by microscopic, multi-pathway protein folding transitions [79–81] (figure S1).” This approach is also reminiscent of evolutionary analyses of adaptive landscapes. Is this analogy not appropriate? Some others are mentioned later on and are described as not being good analogies, but it's not really clear to me why that is. Some explanation is warranted.

The original inspiration for our potential energy landscape approach was free energy landscapes of protein folding. We briefly mentioned this in Lines 129-131: “Here, we use an approach of potential energy landscape, directly inspired by the free energy landscape approach to modeling microscopic, multi-pathway protein folding transitions [63–65] (figure S1).”

This is elaborated in our 2020 PNAS paper:

“In the field of protein folding, adopting a statistical physics view and using an energy landscape approach led researchers to recognize that proteins fold via multiple pathways and understand the physical principles (36–38). These near-equilibrium, microscopic systems statistically transition from higher to lower energy states (local minima) on a free energy landscape (increasing thermodynamic favorability). Thermal fluctuation helps the system stochastically cross energy barriers at transition states (saddle points between local minimum basins).”

“(1) The self-propelled system’s state is attracted to a local minimum basin on a potential energy landscape; locomotor transition from one mode to another can be viewed as the system state escaping from one basin and settling into another. (What governs transition?) (2) When it is comparable to the potential barrier, kinetic energy fluctuation from oscillatory self-propulsion helps the system escape from a landscape basin to make locomotor transitions. (When does transition happen?) (3) Escape from a basin is more likely towards a direction along which the escape barrier is lower. (How does transition happen?)”

Both our free energy landscapes and potential landscapes are derived from first principles and based on physically measurable potential energy vary with the system’s physical variables. The behavior of the system on these landscapes follows similar physical principles elaborated above because of this similarity.

There is a loose analogy between our potential energy landscapes and adaptive landscapes that model evolution [93]. However, fitness as defined in adaptive landscapes is not a physically measurable quantity and could be subjective and ambiguous; often, the quantitative relationship between fitness and phenotypic traits are not apparent or not known. Thus, we do not consider it the best analogy.

For other approaches discussed in Section 5b, we did not intend to state that they are not good analogies, but that they differ from our approach in that they do not capture physical interaction from first principles. The potential energy functions used in the cited studies are either fitted or inferred based on previous measurements or assumed to be of certain form without considering the interaction physics.

Revised.

Lines: 150-154

“We emphasize that our potential energy landscapes directly result from physical interaction and are based on first principles, unlike artificially defined potential functions to explain walk-to-run transition [69] and other non-equilibrium biological phase transitions [70], metabolic energy landscapes inferred from oxygen consumption measurements to explain behavioral switching of locomotor modes [18], and artificial potential fields for robot obstacle avoidance [71].

Many of the analyses and discussion in this manuscript assume that the locomotor landscape is horizontal, but I don’t remember if that qualification is made anywhere.

We considered obstacles on flat surfaces, but our energy landscape approach can be extended to obstacles on sloped or curved surfaces as well, by accounting for how the potential energy changes with the surface topology.

Lines 183-184:

“Although our model systems are on horizontal surfaces, terrain interactions on sloped surfaces can also be analyzed using our approach ”

The word ‘surprisingly’ is used in many places in the manuscript (e.g., line 24). For the most part, I didn’t understand what was surprising in these situations. If that word is to be retained, some explanation of why that thing is surprising is warranted.

The physical principles for macroscopic, far-from-equilibrium animals and robots making locomotor transitions are similar to those of microscopic, near-equilibrium protein-folding transitions on free energy landscapes. This was unexpected, given the differences in scale and nature of interaction (mechanical interaction in locomotion vs. van der Waals, dipole, hydrogen bond, hydrophobic, etc. in protein folding).

We removed all uses except the following instance:

Lines: 327-329

“Our empirically discovered physical principles of locomotor transitions are surprisingly similar to those of microscopic multi-pathway protein folding transitions where predictive energy landscape theories have been successful [63–65] (see detail in [44]).”

The supplementary movie is interesting, but it is quite hard to follow because so many things happen at once. Perhaps break up some of the movement in some way, either temporally or by isolating spatially.

Revised to show locomotor modes and landscapes of each obstacle, one at a time.

Specific comments by line number:

25 “General principles of locomotor transitions begin to emerge for this potential energy landscape dominated regime”

Can you provide some of the major examples, even briefly?

Revised.

Lines 17-18:

Across our model systems, we discovered that locomotor transitions are barrier-crossing transitions on a potential energy landscape.

36 “Despite this, far more...”

What does ‘this’ refer to? Please specify to reduce ambiguity. This construction is used in multiple places in the manuscript.

Fixed.

Lines 257:

“This transition is further facilitated by reduced sprawling and differential use of hind legs (figure 4c, iv) [45], which presumably destabilize the system state towards the roll basin.”

Other instances have been removed.

39 “Previous studies have begun to reveal how terrestrial animals stochastically transition across locomotor modes in complex environments.”

How does an animal transition in a stochastic manner? Please explain.

Animal’s transition is stochastic in the sense that during terrain interaction, there is large variation in quantities such as initial and final pose, time spent in each mode before transition, probability of transitioning from one mode to other, sequence of locomotor transitions, total time and number of transitions required for traversal/self-righting, etc.

We mention this briefly in line 189-190:

“For all model systems, we found that the animal’s transitions between modes occur stochastically, with large trial-to-trial variation [44–48,52–54]”

57 Traversing complex 3D what?

Fixed. “... traversing complex 3D terrain ...”.
Line 49.

84 The references should refer to ‘design and control strategies’, not design and control.

Fixed, Line 74.

86 “Inspired by these recent successes, our group has been creating terradynamics of locomotor transitions in complex 3-D terrain”

This usage is puzzling — how exactly has the group been creating terradynamics?

Terradynamics is the study of physical interaction during locomotion in complex terrain (Li, Zhang, Goldman, 2013 *Science*; Li, Pullin, Haldane, Lam, Fearing, Full, 2015, *Bioinspiration & Biomimetics*). We integrate animal and robot experiments to empirically observe the interaction and resulting locomotor transitions in complex 3-D terrain. We then create physics models such as the potential energy landscapes and dynamical templates to understand how locomotor transitions emerge from physical interaction. These are the subject of this review and elaborated throughout the paper.

119 Change to “in fluids) for each model”.

Removed.

137 “To study these, we developed a novel terrain treadmill with modular large obstacles (figure 3d) which records the animal motion at high spatial resolution [64].”

The treadmill records at high spatial resolution?

Clarified.

Lines: 108 to 110:

“To study these, we developed a terrain treadmill with modular large obstacles (figure 3e) to observe locomotion over a long time and distance [55], with which finer features such as antenna and leg motion can be recorded at a higher spatial resolution [56]”

164 “In addition, terrain interaction of robotic physical models are governed by real physics (i.e., “you get the physics for free” [68]).”

The meaning of the quote in parentheses is not obvious — please explain what it means.

Revised.

Lines: 117 to 118:

“Finally, robots as physical models “can’t violate the laws of physics because robots are enacting, not modelling, the laws of physics” [57]).

179 “Locomotor transitions in obstacle traversal and self-righting occur probabilistically via multiple pathways [58].”

Similar to the previous comment, what does it mean that they occur probabilistically? It’s not explained in this paragraph. `

Animal’s transition is stochastic in the sense that during terrain interaction, there is large variation in quantities such as initial and final pose, time spent in each mode before transition, probability of transitioning from one mode to other, sequence of locomotor transitions, total time and number of transitions required for traversal/self-righting, etc.

Revised

Lines 189-190:

“For all model systems, we found that the animal’s transitions between modes occur stochastically, with large trial-to-trial variation [44–48,52–54]”

192 Change leads to lead.

Fixed.

223 “First, we developed dynamic templates”

I believe this phrase should be ‘dynamical templates’ or ‘templates of dynamics’.

Changed to ‘dynamical templates’ in Line 162.

231 “which landscapes cannot yet do due to the lack of system dynamics.”

Do you mean due to the lack of systems dynamical analyses?

Deleted. This is explained in the previous paragraph.

Lines 155-158:

“Our potential energy landscapes do not yet capture system dynamics, which is required for quantitative predictions of locomotor transitions (Section 5a). Despite this limitation, they provided substantial insight into the general principles and strategies of obstacle traversal and strenuous ground self-righting (Section 4a-g).”

233 “In addition, we developed multibody dynamics simulations of robotic physical model [63] to test the effect of randomness in wing-leg phase, which was not possible in the robotic model enabled larger- scale and finer variation of relevant parameters identified from animal and/or robot experiments for more in-depth measurements and dynamic analysis and revealed nuanced effects that may not be testable in robotic physical models like controlled variation of randomness.”

I don't understand this long sentence.

We have split this sentence into multiple ones as below.

Lines 168-172:

“In addition, for strenuous ground self-righting, we developed multibody dynamics simulations of the robot [54] to study the effect of randomness in wing-leg coordination (figure 3m). These simulations enabled large-scale variation of relevant parameters identified from experiments for in-depth analysis, revealing nuanced effects difficult to observe in animals and robots. Finally, simulation is faster than experiments [54].”

250 Change has to have.

Fixed.

256 “which breaks continuous frictional contact and makes the system statically unstable.”

Is this always true? For example, what about insects that always have 3 legs in contact with the ground, or turtles when walking?

This is not always true, but in the context of large obstacle traversal and self-righting being discussed here, it is true. We feel this is clear, as this phrase directly follows “This is because self-propulsion induces continual collisions during obstacle interaction and self-righting”
See lines 195-197.

285 “However, outside of this regime, energy landscape modeling is not useful—for example, not for ballistic jumping over small obstacles with kinetic energy far exceeding potential energy barriers.”

Interesting point, but tell us more — what are other real-world specific examples?

Revised:

Lines 234-237:

“Such examples include: ballistic jumping over small obstacles with kinetic energy far exceeding potential energy barriers, transition from horizontal running to vertical wall climbing [73] with potential energy increasing monotonically, and traversing obstacles that are much smaller or larger than body size”

308 Change provide to provides.

Removed.

314 Change mode to modes.

Fixed (Line 266).

315 Change to robots to go into.

Fixed (Line 267).

325 Change constrains to constrain.

Fixed (Line 286).

329 “We believe that this physical constraint not only has played a role in the evolution of animal morphology, physiology, and behavior for large obstacle traversal”

Do you have evidence of this statement of belief, or is it meant to be posed as a hypothesis?

Revised.

Lines 287-290:

“We suspect that this physical constraint plays a role in the evolution of animal morphology and behavior. This is plausible because morphological [75–77] and behavioral [78] adaptations that facilitate obstacle traversal and self-righting are common when movement is physically constrained by microhabitat properties.”

391 “New energy landscape theories of multi-pathway locomotor transitions will help expand statistical physics to macroscopic, far-from-equilibrium, self-propelled (active) systems [76–78] and help establish physics of living systems at the organismal level.”

What does ‘establish’ mean here? The implication is that physics of living systems at the organismal level is not a recognized area. I would disagree with that contention. For one, there is a recognized NSF program of the same name that has funded organismal projects for years.

We meant that it is still being established as many more model systems and problems (such as how locomotor transitions emerge from physical interaction here) at the organismal level are being added. In addition, our energy landscape approach is the beginning of a statistical physics approach to understand locomotor transition in macroscopic, self-propelled animals and robots. However, to avoid confusion, we replaced “establish” with “expand”.

Lines 335-337:

“Such new theories will help expand physics of living systems to the organismal level and expand statistical physics to macroscopic, far-from-equilibrium, self-propelled (active) systems [61,62]”

429 “It can then plan its strategies in one- shot”

I don't know what this means.

Deleted.

445 Add a period after the citation.

Fixed.

483 Capitalize the book title, here and in other references.

Fixed.

569 Italicize the species name.

Fixed.

532 As a journal article, the title should not be capitalized, here and in other references.

Fixed.

575, 577 Correct the numerical information.

Fixed.

603 Is this reference in review or in press?

This reference is now in press.

Fixed.

696 Capitalize and italicize the genus name, here and in other references.

Removed.

Supplement

28 A ‘bridge’ implies that the animals body fully subtends the gap, but animals cross gaps in manners relevant to this manuscript in ways that go beyond bridging. ‘Gap crossing’ is a more general and appropriate term. See Graham 2020. J. Exp. Zool. A: 333(1):60-73.

Replaced.

Appendix B

We thank Reviewer 3 for further suggestions and have revised the text accordingly.

We have carefully proofread the paper and made edits throughout to improve clarity.

We also moved the more speculative Fig. 5 to supplementary information (now Fig. S8).

Associate Editor:

Board Member: 1

Comments to Author:

The article has been nicely revised. Well done. One reviewer still has some small and useful suggestions to improve readability. Please address those for the camera-ready version.

Reviewer(s)' Comments to Author:

Referee: 1

Comments to the Author(s)

The authors did a good job addressing the comments. I recommend this paper to be accepted.

Referee: 2

Comments to the Author(s)

The manuscript is well revised based on the reviewers comments.

Referee: 3

Comments to the Author(s)

The authors have done a good job of addressing the concerns of the reviewers. I don't have any further major comments. Below are a few remaining small considerations and suggestions that may help to improve the manuscript prior to publication.

Comment from the first submission:

The word 'surprisingly' is used in many places in the manuscript (e.g., line 24). For the most part, I didn't understand what was surprising in these situations. If that word is to be retained, some explanation of why that thing is surprising is warranted.

Author response:

The physical principles for macroscopic, far-from-equilibrium animals and robots making locomotor transitions are similar to those of microscopic, near-equilibrium protein-folding transitions on free energy landscapes. This was unexpected, given the differences in scale and nature of interaction (mechanical interaction in locomotion vs. van der Waals, dipole, hydrogen bond, hydrophobic, etc. in protein folding). We removed all uses except the following instance:

Lines: 327-329^[17]_{SEP} "Our empirically discovered physical principles of locomotor transitions are surprisingly similar to those of microscopic multi-pathway protein folding transitions where predictive energy landscape theories have been successful [63–65] (see detail in [44])."

Now that I have a better understanding of the authors' perspective, I still think this instance needs additional text. It would be much stronger if the reasons for finding it surprising are provided, rather than stating it baldly. Simply add the explanation given in the response above, and then it makes sense.

Revised.

Lines 288-293

The empirically discovered physical principles of locomotor transitions using feedforward self-propulsion (Section 4d) are surprisingly similar to those of microscopic multi-pathway protein folding transitions (see detail in [48]), where predictive free energy landscape theories have been successful [65–67]. This was unexpected, given the differences in scale and nature of interaction (macroscopic contact forces in locomotion vs. ionic and dipole interactions, hydrogen bond, van der Waals force, hydrophobic interaction in protein folding) [66].

Comment from first submission:

This usage is puzzling — how exactly has the group been creating terradynamics?

Author response:

Terradynamics is the study of physical interaction during locomotion in complex terrain (Li, Zhang, Goldman, 2013 Science; Li, Pullin, Haldane, Lam, Fearing, Full, 2015, Bioinspiration & Biomimetics). We integrate animal and robot experiments to empirically observe the interaction and resulting locomotor transitions in complex 3-D terrain. We then create physics models such as the potential energy landscapes and dynamical templates to understand how locomotor transitions emerge from physical interaction. These are the subject of this review and elaborated throughout the paper.

I think that the authors misunderstood my comment. I wasn't looking for an explanation of terradynamics. As stated in this response, terradynamics is an area of inquiry. The current sentence in line 65 is "A physics-based approach by creating terradynamics [20] holds promise for filling this major gap." This sentence doesn't make sense to me. Substitute in other areas of inquiry, say "ecology" or "functional morphology" or "thermodynamics", and it similarly doesn't make sense. "Create" is the wrong word; I think "using" was intended. ^[L]_[SEP]

Revised

Lines 14-16:

Here, we review our progress towards filling this gap by discovering terradynamic principles of locomotor-terrain interaction, using simplified model systems representing distinct challenges in complex 3-D terrain.

Lines 58-59:

A physics-based approach by creating a new field of terradynamics [21] holds promise for filling this major gap.

Lines 69-71:

Inspired by these successes, our group has been expanding the field of terradynamics to locomotor transitions in complex 3-D terrain, by integrating biological experiments, robotic physical modeling, and physics modeling (figure 2).

Other suggestions by line:

20 Change to “understanding have advanced”.

Revised.

Line 21 – 22:

General physical principles and strategies from our systematic studies already advanced robot performance in simple model systems.

29 What does “this” refer to?

Revised.

Lines 30-32

Despite this multimodality, most mechanistic understanding of terrestrial locomotion has been on how animals generate [5–8] and stabilize [9–11] steady-state, limit-cycle-like locomotion using a single mode.

33 Change to “animal behaviors”

“Animal behavior” can be and is often used as a plural.

A Google scholar search returns ~4M results for singular and 2.7M for plural.

Thus, we prefer to keep it singular here.

59 Change to “near-flat”

Fixed.

69 Change to “dynamics techniques”

Fixed.

108 What does “these” refer to?

We revised this paragraph to make it more clear.

Lines 96-99:

To study continual transitions, we developed a terrain treadmill (figure 3e) to study locomotion through large obstacles over a long time and a large distance [58], while allowing finer features such as antenna and leg motion to be observed at high spatial resolution [59]. This research direction is still at an early stage.

116 Change to “allows us” or “allows one”

Revised.

Lines 104-105:

Moreover, running the robot in open loop allows isolating the effects of passive mechanics from that of sensory feedback.

271 What does “this” refer to?

Revised.

Lines 257-260:

This stereotypy is because the potential energy landscape is highly rugged, with distinct basins separated by barriers, and the system is strongly attracted to landscape basins in the potential energy landscape-dominated regime.

275 Change to “landscape”

Removed.

310 Change to “and do not”

Fixed. See caption of figure S8

357 What is “it”? What does “This” refer to?

Revised.

Lines 333-336:

If future potential energy landscape theories can predict how locomotor performance depends on relevant system parameters (Section 4d-g), they will provide a proxy for fitness landscapes [91]. Such proxy fitness landscapes will reveal how locomotor fitness exerts selective pressure on morphology and behavior that affect locomotor transitions via physical interaction.

376 Pull out the long parenthetical and make it into a sentence that follows, as in “For instance, dense forests...”

Removed this sentence.